



# How accurately do engineering methods capture floating wind turbine performance and wake? A multi-fidelity perspective

Stefano Cioni[1], Francesco Papi[1], Pier Francesco Melani[1], Alessandro Fontanella[2], Agnese Firpo[3], Andrea Giuseppe Sanvito[3], Giacomo Persico[3], Vincenzo Dossena[3], Sara Muggiasca[2], Marco Belloli[2], Alessandro Bianchini[1]

[1]Department of Industrial Engineering, Università degli Studi di Firenze, Firenze, Italy
[2]Department of Mechanical Engineering, Politecnico di Milano, Milano, Italy
[3]Department of Energy, Politecnico di Milano, Milano, Italy

*Correspondence to*: Alessandro Bianchini (alessandro.bianchini@unifi.it)

**Abstract.**

Despite an increasing number of experimental and numerical studies, the influence of platform motion on wake dynamics (wake recovery and turbulence production) in floating offshore wind turbines is still an open research question. In particular, efforts are being made to understand the accuracy of numerical models in use so far for fixed-bottom turbines when they are

applied to floating configurations. Similarly to what has been done in IEA's OC6 task, in this work a multi-fidelity approach is leveraged to investigate the capabilities of engineering models to capture the wake dynamics of a wind turbine model under imposed motion. Differently from previous studies, however, many more different operating conditions have investigated, including surge, pitch, yaw and wind-wave misalignment cases; moreover, numerical methos are here consistently applied to the same test cases, which are part of the first experimental round of the NETTUNO project. More specifically, Free Vortex

Wake (FVW), Actuator Line Model (ALM) and blade resolved CFD simulations have been benchmarked and their capabilities in predicting the mean wake response and the onset of velocity oscillations in the wake of a floating wind turbine were evaluated. Results showed that, up to 5D and in the operating conditions tested, platform motion has limited impact on the wake in terms of wake deficit. However, significant velocity oscillations are observed at a platform reduced frequency of 0.6 which could be detrimental for downstream machines. An investigation of the vortex structures in the wake showed that these

velocity oscillations might be caused by the interaction of vortex structures generated under sinusoidal platform motion rather than by unsteady aerodynamic response of the rotor. FVW methods, if properly tuned, can correctly capture the wake response up to 3D from the rotor, but to simulate the wake response up to 5D, higher-fidelity methods are required. Significant improvements are achieved with ALM CFD simulations, even though an URANS approach might struggle to correctly predict the wake dissipation due to the interaction between the free-stream turbulence and wake.



## 1 Introduction

Floating offshore wind turbines (FOWTs) have been one of the key study areas for wind energy research in the last few years because they represent the most promising way of exploiting the vast wind energy potential in deep waters. Despite recent research efforts, further work is required to improve the understanding of the complex interactions taking place between wind-driven loads, the aero-servo-elastic behavior of the rotor, and the hydrodynamics of the floater (Veers et al., 2022).

In fact, the additional degrees of freedom alter the dynamic behavior of the system, both in terms of aerodynamic response of the rotor, dynamics of the wake and control system. In particular, from an aerodynamic point of view, the effect of platform motion is two-fold. First, the flow field around the rotor is modified (Chen et al., 2020; Sebastian and Lackner, 2013; Tran and Kim, 2015), causing, for example, local differences in the relative wind speed. Second, during the complex motion of the platform, especially in case of severe sea states, the blades might enter their own wake, affecting the local induction and wake behavior (Dong and Viré, 2022; Papi et al., 2023; Ramos-García et al., 2022).

Understanding the wake dynamics of a floating wind turbine represents also a topic of interest, as to reduce the installation and O&M costs, floating wind turbines will be installed in clusters, leading to significant wake interactions.

In fact, analogously to fixed-bottom machines, floating wind turbines placed downstream operate in the wake of the upstream machines, leading to reduced power production and increased loading. For this reason, installing wind turbines in clusters represents a non-trivial issue and requires careful design. Research has focused on improving the understanding of wake dynamics to provide industry with valuable information and models which may aid the optimization of future wind farms.

Despite recent progress in the investigation of fixed-bottom wind turbine wakes (Porté-Agel et al., 2020), understanding the wake dynamics of FOWTs and the differences from the fixed-bottom behavior over a range of operating conditions remains an open question. In fact, it is still unclear how the additional DOF of FOWTs impact the wake development.

Previous work has shown that platform motion induces periodic oscillations in aerodynamic rotor loading due to the changes in relative velocity (Bergua et al., 2022; Fontanella et al., 2021; Schulz et al., 2023; Taruffi et al., 2024) and this is expected to impact also the wake dynamics. At first, it was assumed that the oscillations in rotor aerodynamic response could lead to improve mixing in the wake, improving the velocity recovery. This would lead to better performance of floating wind turbines in a waked configuration, due to the reduced velocity deficit and increased energy available in the wake.

However, preliminary results concerning wake deficit in the wake have shown conflicting results. Wind tunnel experiments performed by Fontanella et al (2021) during the UNAFLOW campaign and subsequently during the NETTUNO experimental campaign, have shown that the investigated sinusoidal platform motion conditions, including surge, pitch yaw and wind-wave misalignment, have only minor impact on wake deficit up to 5D from the rotor. In contrast, Messmer et al (2024a, b), investigated the wake response for a distance up to 8D and observed further differences in the average wake recovery, especially when the inflow turbulent intensity is below 3%. At higher turbulence levels, the first full scale LIDAR measurements of a 6MW floating wind turbine have also shown no significant difference in wake recovery (Angelou et al., 2023). Indeed, numerical simulations performed by Pagamonci et al. (2025) have shown that inflow turbulence can have a



significant impact on wake development of a floating wind turbine. Numerical studies have also investigated the wake deficit of floating wind turbines under imposed motion (Arabgolarcheh et al., 2022; Kleine et al., 2022; Ramos-García et al., 2022).

Results have shown that the additional DOF can lead to faster breakdown of the vortex structures in the near wake (Arabgolarcheh et al., 2022; Kleine et al., 2022), due to periodic variations in the position (Tran and Kim, 2015) and intensity of the tip vortices (Cioni et al., 2023). Nevertheless, the impact on the wake deficit was limited in the investigated conditions (Pagamonci et al., 2025; Ramos-García et al., 2022).

In addition to possible differences in wake deficit, the platform motion could also induce low-frequency oscillations in the

wake (Kleine et al., 2021), which could lead to increased loading on downstream turbines or even excite the response of the low-frequency platform modes (Veers et al., 2022). Wind tunnel measurements performed by Fontanella et al. (2022) showed that surge and pitching motions of a FOWT can induce velocity oscillations in the wake that are propagated downstream at a characteristic speed. Such oscillations are maximized for specific oscillation frequencies (Fontanella et al., 2024; Kleine et al., 2022; Schliffke et al., 2024), resulting in large streamwise velocity oscillations.

These large coherent structures are generated in the wake even for turbulent inflows, which could lead to increased loading on downstream turbines (Duan et al., 2025; Messmer et al., 2024b; Pagamonci et al., 2025; Schliffke et al., 2020). However, the current numerical methods may struggle to correctly capture their amplitude, as shown in the code-to-code comparison performed during the OC6 phase III project.

A second issue hampering the development of floating wind turbines is, in fact, validation. The vast availability of numerical

methods has allowed industry and academia to study the aerodynamic and wake response of floating wind turbines without the need for expensive models or prototypes. These methods have been developed and extensively tested for fixed-bottom wind turbines; however, similar validation campaigns have still not yet been performed for floating wind turbines, mainly due to the lack of open-acess experimental or field data. For this reason, simulation codes may not always provide reliable results, and results may depend on the specific numerical setup employed. For example, lower fidelity methods are often significantly

affected by tuning parameters, which may alter simulation results drastically.

In this framework, the NETTUNO project aims to evaluate the reliability of the available numerical codes, as well as shed further light into the actual physics governing wake dynamics of a FOWT employing a multi-fidelity approach. The analysis was carried out on a scaled model of the DTU 10MW, which, with a diameter of 2.38, represents the largest model of a floating wind turbine investigated to date. In this way, this work builds up on previous studies on floating wind turbines, which usually

employ a smaller scale model (Messmer et al., 2024a) or a porous disc (Schliffke et al., 2024). The results obtained from Free Vortex Wake (FVW) and ALM URANS simulations were compared with the experiments to evaluate their capabilities and limitations in predicting the rotor aerodynamic response and wake dynamics. Following the comparison the FVW method was properly tuned to improve its reliability. For the ALM simulations, two different ALM models and numerical setups employed by two different institutions, namely Universtiy of Florence and Politecnico di Milano, have been employed to provide a range

of accuracy for this methodology and sensitivity to simulation parameters. The validation of the numerical models was carried out over a significant range of operating conditions, including yaw motion and wind-wave misalignment, up to 5D from the





rotor, representing the most in-depth evaluation of lower-fidelity models to date. In fact, previous works have focused mainly on surge motion, while an in-depth comparison under more complex cases is still lacking. Additionally, previous multi-fidelity analysis, such as those carried out during the OC6 phase III project, were limited to up 2.4D, allowing for a limited investigation of the wake dynamics.

The numerical models were also compared with high-fidelity blade-resolved URANS simulations, which provide complementary data to the experiment in terms of spanwise rotor loads. This further comparison sheds light on the rotor aerodynamic response and provide a further benchmark for the lifting line models. Additionally, 2D velocity fields from ALM LES simulations performed during the NETTUNO project (Pagamonci et al., 2025) , are leveraged to investigate whether lower-fidelity models can capture the coherent structures generated in the wake due to platform motion. Despite its significant computational cost, this method should better capture the interaction of the wake with free stream turbulence and the development and collapse of the vortex structures in the wake.

This work is structured as follows: Sect. 2 and Sect. 3 describe the experimental setup and numerical methods employed in this work, respectively. The main outcomes of the analysis are presented in Sect. 3 and 4, where the aerodynamic loads and wake response are evaluated. Finally, the main conclusions of this work are summarized in Sect. 5.

## 2 Experimental data for benchmarking

The experimental data employed as benchmark in the NETTUNO project was obtained during an experimental campaign carried out at the wind tunnel of Politecnico di Milano (Fontanella et al., 2024). A 1:6 scaled model of the DTU 10MW (Bak et al., 2013) turbine was placed on a 6DOF robot and tested under multiple imposed platform motion conditions. The main geometric parameters of the wind turbine model are summarized in Table 1. Sinusoidal surge, pitch and yaw motions were tested with varying amplitudes and frequencies, to provide the most complete database to date.

The wind tunnel campaign focused on surge and pitch motions of the platform. The wake development was assessed with varying amplitudes, frequencies of motion, and apparent wind speed at the hub, which ranged from 0.1 ms$^{-1}$ to 0.7 ms$^{-1}$. Following existing literature (Bergua et al., 2022; Fontanella et al., 2021; Taruffi et al., 2024), the investigated frequencies are expressed in form of rotor reduced frequency, as in Eq. 1:

$$f_r = \frac{fD}{U}, \tag{1}$$

where f is the platform motion frequency, D is the rotor diameter, and U the free-stream velocity. According to Eq. 1, the investigated range between 0.5 to 2 Hz corresponds to a range of reduced frequencies between 0.3 and 1.2. These frequencies are considered representative of full scale turbines. In particular, the reduced frequencies of 0.3 and 0.6 correspond to full-scale of 0.02 and 0.04 Hz, typical of rigid-body modes, and the reduced frequency of 1.2 corresponds to a full scale frequency of 0.08, representative of wave-induced motion. A summary of the tested conditions is shown in Table 2.





Additional tests were performed for a sinusoidal yaw motion of the platform, to investigate whether such platform motion conditions may cause differences in wake development. In this case the analysis was carried out for a fixed amplitude of yaw motion, which is considered representative of operating conditions for floating wind turbines. The wake behaviour was

measured as a function of the frequency of platform motion.

In the surge, pitch, and yaw cases, the wind direction was aligned with platform motion, mimicking the common situation in which waves and wind are aligned along the same direction. However, a misalignment, $\gamma$, between platform motion and wind direction may arise in some cases. To evaluate the impact of such conditions on wake development, combinations of surge and sway, and pitch and roll motions were tested. During these tests, the amplitude and frequency of motion were maintained

constant, and the impact of the misalignment angle was evaluated between 15° and 45°.

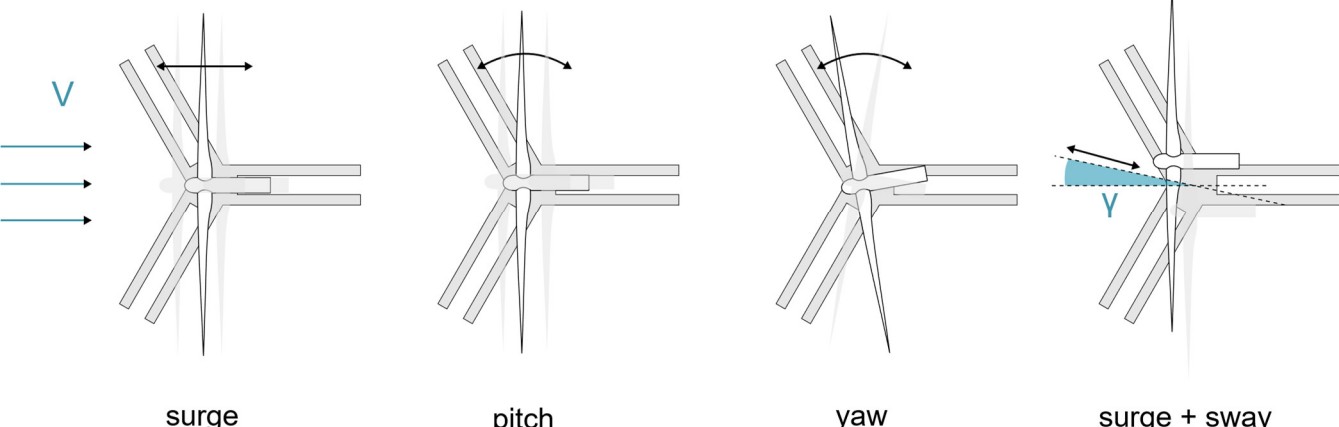

**Figure 1 Summary of platform motion conditions investigated during the NETTUNO project.**

Only sinusoidal motions of the platform were investigated, as the objective of this work is improving the current understanding of the physics governing the wake development of a floating wind turbines and not provide a representation of realistic wake

dynamics in the field. A more realistic motion of the floating substructure could be imposed through a combination of all the 6 DOFs, but may lead to different wake dynamics to those during these model scale tests.

During the tests the tower top loads and wake response were measured. A Hot Wire Anemometer (HWA) was used to measure horizontal and vertical velocity profiles at 3D and 5D from the rotor, using a traversing system. The vertical traverses were acquired for some of the operating conditions to evaluate if the motion induces an a-symmetric response in the wake. The

inflow conditions were maintained constant at 4 m/s, with an inflow turbulence level of about 1.5% over the rotor area. The rotational speed of the turbine was kept constant at 240 rpm. Additional details about the experimental set up are found in Fontanella et al. and the complete dataset is available open-access (https://doi.org/10.5281/zenodo.13994979).



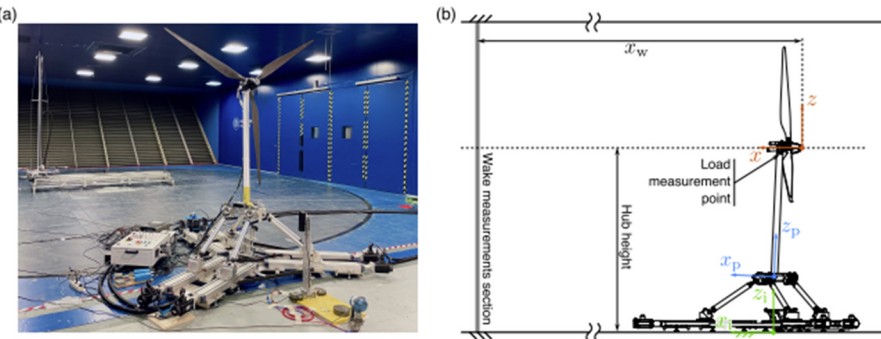

**Figure 2 Experimental set up during the NETTUNO wind tunnel campaign. (a) Wind turbine model and traversing system. (b) Sketch of wind turbine model and reference systems. Image from (Fontanella et al., 2024).**

**Table 1 Summary of wind turbine model main parameters.**

| Parameter | Value | Unit | Parameter m | Value | Unit |
|---|---|---|---|---|---|
| Rotor diameter (D) | 2.381 | m | Shaft tilt angle | 5 | ° |
| Blade length | 1.102 | m | Tower-to-shaft distance | 0.064 | m |
| Hub diameter | 0.178 | m | Tower length | 1.400 | m |
| Hub height | 2.190 | m | Tower diameter | 0.075 | m |
| Rotor overhang | 0.139 | m | Tower base offset | 0.730 | m |

## 3 Numerical methods

The test matrix described in Sect. 2.1 was simulated employing two different numerical approaches: the FVW method implemented in QBlade (Marten, 2020) and a CFD ALM model implemented in CONVERGE (2025). The details of each simulation setup are described in Sect. 3.1 and 3.2. Additional ALM simulations of some surge and pitch cases were performed in OpenFOAM employing the ALM code developed by Sanvito et al. (2024). These simulations are included in this work to showcase how different numerical setups and tuning of an ALM simulation might impact its reliability. As a further benchmark for the lower-fidelity methods, blade-resolved URANS CFD simulations of the platform pitching case were performed in Ansys Fluent, at a frequency of 2 Hz and an amplitude of 0.3°. This case also corresponds to the highest frequency pitching motion investigated during the OC6 phase III project, where the most significant differences between simulations and experiments were observed. The blade-resolved simulation was used to shed further light into the spanwise aerodynamic response of the rotor and providing further validation of the low fidelity methods. The numerical setup is described in Sect. 3.3. Finally, the ALM LES results obtained by Pagamonci et al. (2025) for the pitch motion case, characterized by an amplitude of 1.3° and a reduced frequency of 0.6, are included as benchmark for the analysis of the wake dynamics. In fact, this high-fidelity method can provide 2D velocity fields without the need of complex PIV measurements. A summary of the operating conditions investigated in this work is shown in Table 2.





**Table 2 Summary of operating conditions investigated in this work. A is the amplitude of platform motion, $f$ and $f_r$ are the frequency and rotor reduced frequency, respectively, and $\gamma$ is the misalignment angle.**

| Platform motion | $A$ [m] or [°] | $f$ [Hz] | $f_r$ [−] | $\gamma$ [°] | FVW | ALM-URANS UNIFI | ALM-URANS POLIMI | ALM-LES | BR-URANS |
|---|---|---|---|---|---|---|---|---|---|
| Fixed | - | - | - | - | ✓ | ✓ | ✓ | ✓ | |
| Surge | 0.032 | 0.5 | 0.3 | 0 | ✓ | ✓ | ✓ | | |
| | 0.064 | 0.5 | 0.3 | 0 | ✓ | ✓ | | | |
| | 0.016 | 1 | 0.6 | 0 | ✓ | ✓ | | | |
| | 0.032 | 1 | 0.6 | 0 | ✓ | ✓ | ✓ | | |
| | 0.048 | 1 | 0.6 | 0 | ✓ | ✓ | | | |
| | 0.016 | 2 | 1.2 | 0 | ✓ | ✓ | | | |
| | 0.032 | 2 | 1.2 | 0 | ✓ | ✓ | ✓ | | |
| | 0.048 | 2 | 1.2 | 0 | ✓ | ✓ | | | |
| | 0.032 | 1 | 0.6 | 15 | ✓ | ✓ | | | |
| | 0.032 | 1 | 0.6 | 30 | ✓ | ✓ | | | |
| | 0.032 | 1 | 0.6 | 45 | ✓ | ✓ | | | |
| Pitch | 1.3 | 0.5 | 0.3 | 0 | ✓ | ✓ | ✓ | | |
| | 2.5 | 0.5 | 0.3 | 0 | ✓ | | | | |
| | 3 | 0.5 | 0.3 | 0 | ✓ | | | | |
| | 0.60 | 1.0 | 0.6 | 0 | ✓ | | | | |
| | 1.30 | 1.0 | 0.6 | 0 | ✓ | ✓ | ✓ | ✓ | |
| | 1.90 | 1.0 | 0.6 | 0 | ✓ | | | | |
| | 0.3 | 2.0 | 1.2 | 0 | ✓ | ✓ | ✓ | | ✓ |
| | 0.60 | 2.0 | 1.2 | 0 | ✓ | | | | |
| | 1.30 | 2.0 | 1.2 | 0 | ✓ | ✓ | ✓ | | |
| Yaw | 2 | 0.5 | 0.3 | 0 | ✓ | ✓ | | | |
| | 2 | 1 | 0.6 | 0 | ✓ | ✓ | | | |
| | 2 | 2 | 1.2 | 0 | ✓ | ✓ | | | |

## 3.1 Free Vortex wake simulations

The same test cases investigated during the wind tunnel campaign were simulated using the lifting line free vortex wake method implemented in QBlade (Marten, 2020), version 2.0.7. In this approach, the rotor aerodynamic behaviour comes from tabulated polar data. The aerodynamic coefficients for lower angles of attack were obtained during wind tunnel tests (Fontanella et al., 2021), extended to 360° using the Viterna method [(Viterna and Janetzke, 1982)], and then corrected to account for rotational





augmentation effects using the method by Du and Selig (1998). The workflow is described in detail in Robertson et al. (2023). The same polar dataset was employed during the OC6 phase III project (Bergua et al., 2022), providing good agreement in terms of aerodynamic loading between lifting line simulations and experiments.

In the simulations, the lifting line is discretized in 30 panels using a cosine distribution. The effect of dynamic stall is considered

with the model proposed by Oye et al. (1991). As the wind tunnel walls could not be included in the simulations, the impact of blockage was evaluated with Glauert's correction and the inflow speed was increased to $4.19\ ms^{-1}$ (Cioni et al., 2023).

Following the analysis carried out during the OC6 phase III project (Bergua et al., 2022; Cioni et al., 2023), it was observed that FVW simulations carried out by different participants led to significant differences in the results, despite employing the same aerofoil polars. For this reason, an analysis of the main simulation parameters was performed to investigate their impact

on the accuracy of the results. In particular, the impact of inflow turbulence and vortex filaments parameters was investigated. During the OC6 project, the participants assumed laminar inflow in the numerical simulations, due to the low inflow turbulence in the wind tunnel of about 2%. However, recent studies showed that even small turbulence levels of about 2% can have a significant effect on wake development of a floating wind turbine (Pagamonci et al., 2025). For this reason, turbulence was introduced in the FVW simulations, to evaluate the impact on the accuracy of the results.

In QBlade turbulence is introduced by defining a turbulent velocity timeseries over an imposed grid. The turbulent inflow spectrum of the Politecnico di Milano wind tunnel, measured during previous experimental campaigns (Fontanella et al., 2021), was used to generate the turbulent inflow imposed in the simulations. The timeseries was defined using the software Turbsim (Jonkman, 2009) in order to match the experimental one.

The introduction of turbulence in the simulation can be performed using different approaches in FVW codes. In QBlade two

options are available to consider the effect of turbulence on the vortex filaments:

- The vortex filaments are convected with the mean hub velocity (MEAN)
- The vortex filaments are convected with the local turbulent velocity (LOC)

The two methodologies were investigated to evaluate the impact on the simulation accuracy. Results are shown in Figure 2.

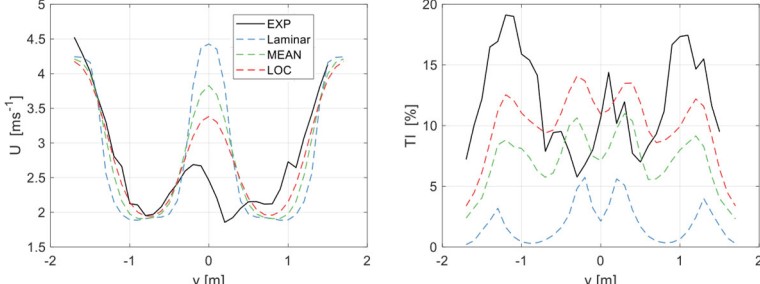

**Figure 3 Impact of different turbulence modelling in FVW simulations in a fixed-bottom configuration.**

The inflow turbulence and its modelling affect the wake results drastically. In terms of average wake profile, the inflow turbulence leads to increased mixing especially at the centre of the wake, showing better agreement with the experimental results in comparison to the laminar inflow conditions. Employing the LOC strategy shows better results, as this methodology can account for the impact of turbulence on the vortex filaments in the wake. Analogous results are observed in terms of wake

turbulence intensity values, as introducing turbulence with the LOC methodology leads to improved agreement with the experimental data.

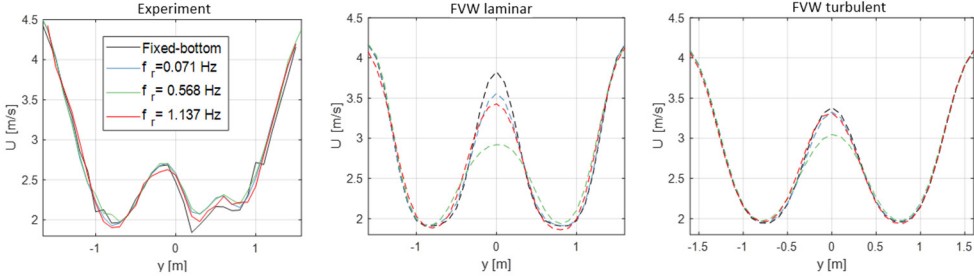

**Figure 4 Impact of turbulence on mean velocity profile at 3D from the rotor for different reduced frequencies of pitching motion. (left) experimental data (middle) FVW results with laminar inflow, (right) FVW results with turbulent inflow.**

The impact of turbulence is even more significant when platform motion is considered (see Figure 4). Indeed, in laminar inflow conditions, a pitching motion of the platform causes improved mixing and faster transition in the central parts of the wake, in comparison to the fixed-bottom value. However, when a 2% inflow turbulence is introduced, the impact of platform motion becomes less significant, in agreement with the results observed during the wind tunnel campaign (see Figure 4, left). This result shows that inflow turbulence cannot be neglected when the wake of a floating wind turbine is investigated and that the

FVW method can correctly account for this, if properly implemented.

Due to the free stream velocity oscillations in the inflow, it is not sufficient to run the simulation until numerical convergence, but the rotor loads, and wake parameters need to be acquired over multiple cycles of platform motion to achieve statistical convergence. The data obtained from the simulations is then phase-averaged over a cycle of platform motion to obtain the mean wake behaviour.

The impact of the main simulation parameters on the simulation results was also tested. In fact, FVW models are are affected by the user-defined initial size of the vortex filaments generated at the lifting line, $r_0$. and by the diffusion properties of the vortex filaments, calculated as

$$V_{ind} = -\frac{1}{4\pi} \int \Gamma \frac{\boldsymbol{r} \times \boldsymbol{dl}}{r^3 + r_c^2} \tag{2}$$

Where the core radius is calculated as

$$r_c = r_0 + \sqrt{\frac{4a\delta_v \nu \Delta t}{1 + \epsilon}} \tag{3}$$

In Eq. (4), $a = 1.25643$ is a constant, $\delta_v$ is the turbulent viscosity coefficient, $\nu$ is the kinematic viscosity, $\Delta t$ the time step size and $\epsilon$ is the strain rate of the vortex filament. The effect of vortex core diffusion is accounted using the vortex turbulent viscosity, $\delta_v$. As this parameter is increased, the dissipation of the vortex filaments is faster (Marten, 2020).





Both the initial core radius and the turbulent vortex core viscosity have a significant effect on wake development. For this

reason, the experimental tests were used as benchmark and a parametric sweep was performed to identify the best combination for improved reliability.

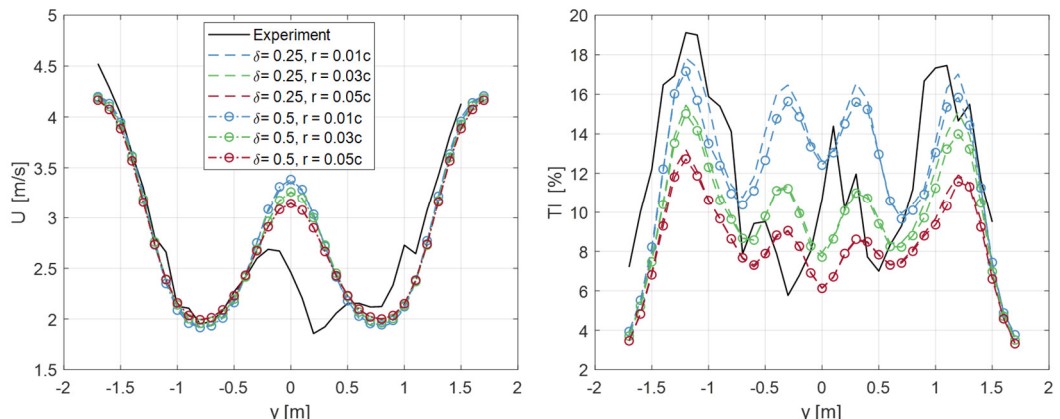

**Figure 5 Impact of vortex viscosity and initial core radius on wake behaviour at a distance of 3 diameters from the rotor.**

The combination of the two parameters affects both the mean velocity and turbulence intensity in the wake, even though the

impact on the latter is more significant. Increasing the initial core radius, the strength of the tip and root vortices decreases, leading to reduced turbulent intensity in the central and outer parts of the wake. Similarly, increasing the vortex viscosity reduces the strength of the vortex structures and spreads out the turbulent intensity gradients. From the parameter sweep, the combination of $\delta_\nu = 0.25$ and $r_0 = 0.03c$ was selected, as it guaranteed the best agreement with the experimental data both in the central and outer parts of the wake.

**2.2 ALM CFD simulations**

The ALM provides an invaluable tool for simulating the wake behaviour of wind turbines. Indeed, this method combines the accuracy of CFD in the wake, with the simplified rotor modelling of a lifting line approach. In this work, the most relevant test cases from Table 2 were simulated using an ALM model and the commercial software CONVERGE v3.1. Some of the surge and pitch operating conditions were also simulated in OpenFOAM using the ALM implemented by Sanvito et al. (2024) to

evaluate the sensitivity of this approach to setup and turning parameters. The two different numerical models have been employed during a blind test, i.e., employing different numerical codes and set ups without prior tuning, to provide a range of predictability of the ALM approach. As the first method was developed within University of Florence (UNIFI) and the second by Politecnico di Milano (POLIMI), the two ALM models and corresponding results will be distinguished by the corresponding institution name in the remaining of this work. The main simulation parameters for the two numerical setups are summarized

in Table 3.





**Table 3 Main setup parameters for the two ALM solvers.**

| *Parameter* | **POLIMI** | **UNIFI** |
|---|---|---|
| *Number of actuator line points* | 75 | 55 |
| *Sampling method* | Vortex method | Line-average method |
| *Regularization kernel* | Piece-wise $\delta$ kernel | 2D axysimmetric gaussian kernel |
| *Size of regularization kernel* | $2.7\Delta$ | $\max(1.6\Delta, 0.25c)$ |
| *Dynamic stall model* | Not employed | Not employed |
| *Tip losses correction* | Not employed | Sorensen tip losses model |

Both ALM simulation setups included the wind tunnel walls in the simulation; UNIFI included in the simulation the nacelle, tower and robot, which are not present in the numerical setup used by POLIMI. To limit the computational effort, slip walls were employed on the wind tunnel ceiling and lateral walls in both numerical setups, and the domain was reduced by 10cm to account for the reduction in cross section due to boundary layer expansion, employing the same setup used during the OC6 phase III (Robertson et al., 2023). The boundary layer on the wind tunnel floor was solved by UNIFI, as the robot was included

in the simulations. In contrast POLIMI further reduced the computational domain by 2 cm on the wind tunnel floor (Robertson et al., 2023) to limit the computational cost, as the robot is not included in the simulation. The numerical domain was discretized in both numerical approaches using a structured grid, with a mesh size of about 0.015m (0.0062D) in the rotor region. UNIFI employed local mesh refinement in the near included additional adaptive mesh refinement in regions characterized by high velocity gradients. In contrast, POLIMI employed a uniform grid both in the rotor region and wake of 0.015m.

For the lifting line model, both ALM simulations employed the same polars of the FVW method, as described Sect. 3.1. In terms of velocity sampling, UNIFI employed the line average approach (Jost et al., 2018), where the velocity is computed as an average over a circle centred on the aerodynamic centre. In this way, the contribution of circulatory effects on the induced velocity cancels out. A total of 20 sampling points was used along a circumference of 1 chord around the aerodynamic centre. In contrast, POLIMI employed the vortex-based method as described in Sanvito et al. (2024).

For the spreading of the aerodynamic loads, UNIFI used a three-dimensional piece-wise $\delta$ kernel with a radius calculated depending on the mesh size, $\Delta$ and local chord value, $c$, as, $\epsilon = \max(0.25c, 1.6\Delta)$ (Xie, 2021). This setup guarantees adequate compromise between numerical stability and avoiding excessive spreading of the rotor force, which would lead to underestimation of the rotor induction. The numerical setup employed here is the same as the one used during the OC6 phase III project (Bergua et al., 2022), as the results showed good agreement with experimental data. In contrast, POLIMI used a 2D

axisymmetric gaussian kernel with a width of $2.7\Delta$, where $\Delta$ is the size of the grid in the rotor region, following the independence study performed in (Sanvito et al., 2024). Both Actuator line models did not include dynamic stall but differences are found in the modelling of tip effects, as UNIFI employed the Sorensen tip losses model (Dağ and Sørensen, 2020), while POLIMI did not use any correction.





For the solution of the Navier-Stokes equations in the computational domain, both UNIFI and POLIMI employed a URANS
approach with the $k - \epsilon$ turbulence model for the closure of the turbulence equations. Additionally, both simulation setups
assumed a compressible flow. The PISO and PIMPLE algorithm were used to solve the fluid equations by UNIFI and POLIMI,
respectively. At the inflow, the turbulence intensity is set to guarantee a value of 2% at the rotor plane , replicating wind tunnel
conditions.

Finally, comparable time steps of the simulation were employed in the simulation setups, equal to $4.16 \cdot 10^{-4}$ and $5 \cdot 10^{-4}$,
for UNIFI and POLIMI, respectively. The size of the time step was chosen in order to guarantee numerical stability of the
ALM model. The ALM simulations were run over multiple cycles of platform motion, until numerical convergence. Since the
URANS methodology only accounts for turbulence through the transport of the turbulent kinetic energy and dissipation rate,
the simulations show only minor differences between successful cycles of platform motion. For this reason, the simulations
were run until numerical convergence of the wake, which required about 15 seconds of simulation time.

In this work, the results from numerical models are benchmarked also with available data from LES simulations performed by
Pagamonci et al. (2025). The same ALM model and geometry are used as the setup employed by UNIFI. Details about the
simulation setup and methodology employed are found in the original publication (Pagamonci et al., 2025). The high-fidelity
LES data is used here to provide detailed 2D representations of the vortex structures in the wake, complementing the
experiment. In fact, similar details would require complex measurements in the wake such as PIV, which are not available at
this time. An in depth comparison of the LES results with the experimental data is out of the scope of this work and is described
in depth in Pagamonci et al. (Pagamonci et al., 2025).

### 2.3 Blade-resolved simulations

Blade-resolved CFD simulations of a pitching case at an amplitude of 0.3° and a frequency of 2Hz were performed to obtain
further information about the spanwise rotor aerodynamic response and provide further benchmark for the lower-fidelity
models. This case was selected as the most significant differences were observed for these operating conditions when
comparing experiments and simulations during the OC6 phase III (Cioni et al., 2023).

The CFD blade-resolved simulations were performed using the software Ansys Fluent (v231). The whole geometry of the
turbine, including the rotor, nacelle, and tower was simulated. The Navier-Stokes equations were solved using a URANS
approach, employing the k-omega SST turbulence model for the closure of the fluid equations, with the gamma-algebraic
boundary layer transition model (Menter et al., 2015).



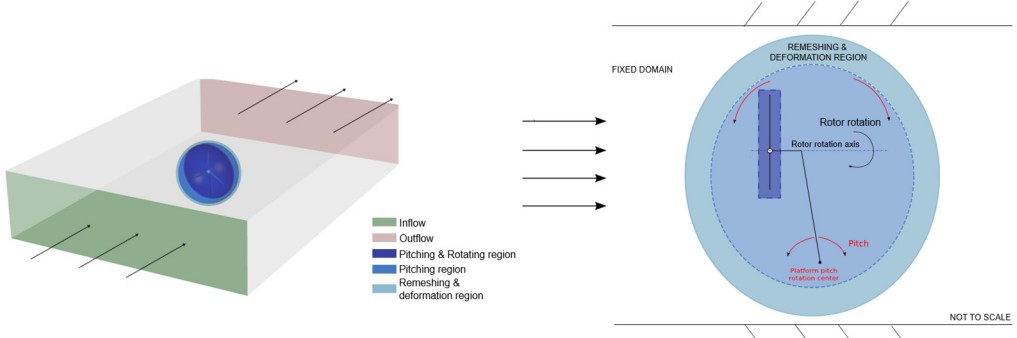

**Figure 6 Computational domain of the Blade-resolved simulation. The inflow and outflow boundaries are highlighted. Additionally, the rotor region, moving region, and remeshing and deformation regions are shown.**

The numerical domain, including the wind tunnel walls, was discretized using a total of about 60 million elements. To limit
the number of elements the size of the domain was reduced by the boundary layer thickness on all the wind tunnel walls and the robot was neglected. In this way, the impact of the boundary layer formed on the wind tunnel walls on blockage is considered without explicitly solving the fluid equations. The rotor region was discretized using polyhedral elements to limit the number of cells and consequently the computational cost of the simulation. In the wake, tetrahedral elements were used, including local refinements in the root and tip regions. In fact, capturing these structures is essential to correctly predict the
rotor aerodynamic response and wake dynamics. A sliding mesh approach was used to include the rotation of the rotor. The pitching motion was imposed to a spherical region including the wind turbine model, while the wind tunnel domain is kept fixed. The relative motion between the two regions is compensated through a spherical shell, where the mesh is deformed and regenerated automatically (Figure 7). This innovative strategy was used to limit the computational cost of the remeshing process. Further details about the rotor and wake discretization and simulation setup are provided in Cioni et al. (2025).
The inflow conditions were set to replicate the experiment, imposing the same mean velocity and turbulent intensity. At the outflow the ambient pressure was imposed. The SIMPLE scheme was used as solution algorithm and the time advancement was performed with a second order upwind approach and a discretization of $4 \cdot 10^{-4} s$, corresponding to a rotor rotation of $0.5°$. The simulation was run for a total of $9s$ to achieve the numerical convergence of both the rotor loads and wake velocities up to 3D.

## 325   3 Multi-fidelity analysis of rotor loads

In this section, the capabilities of the engineering models to capture the aerodynamic response of the rotor was evaluated by comparing the results with available experimental data and spanwise loads obtained from blade-resolved simulations. In this way, the limitations of each method in predicting the rotor aerodynamic response can be evaluated.



### 3.1 Rotor integral loads

The rotor thrust and torque obtained from the wind tunnel campaign and the FVW and CFD ALM simulations were compared across the investigated test cases. Table 4 shows the mean thrust and torque values for the surge and pitch motion cases. The FVW and CFD ALM results show good agreement with the experiment and the small differences are withing the experimental uncertainty of the data. Across all the investigated platform motion cases, the mean thrust and torque values from simulations and experiments show only minor differences, as platform motion only induces periodic oscillations in rotor loads without

affecting the average aerodynamic response across the investigated operating conditions, independently of platform motion amplitude, frequency or type of motion.

**Table 4 Summary of rotor thrust and torque values for surge and pitch motion cases. Results are averaged across the investigated platform motion conditions. Only minor differences were observed in terms of average rotor loads across the different operating conditions.**

|  | *Thrust [N]* | *Torque [Nm]* |
|---|---|---|
| *Experiment* | 36.47 | 2.97 |
| *FVW* | 35.714 | 3.058 |
| *ALM UNIFI* | 35.185 | 3.218 |
| *ALM POLIMI* | 35.665 | 3.158 |


To further compare numerical and experimental results, the amplitude and phase shift of the rotor thrust and torque oscillations was investigated, to evaluate possible differences across the operating conditions. The rotor thrust and torque time series were post processed using the Fourier transform to extract the amplitude and the phase shift with regard to platform motion of the oscillations. In quasi-steady conditions,, i.e., in the absence of significant unsteady effects, platform motion should lead to

periodic oscillations with an amplitude proportional to the reduced frequency and a phase shift of -90 with regard to platform motion (i.e., corresponding to the rotor loads being in phase with the relative velocity profile induced by the motion) (Bergua et al., 2022). For both surge and pitch cases, the amplitude of thrust and torque oscillations shows a linear trend with the reduced frequency, and the slope of this linear relationship increases with the amplitude of motion. The numerical results predict with only minor differences the amplitude of the loads oscillations until a reduced frequency of 0.6. Some differences

arise at the reduced frequency of 1.2, where the numerical methods underpredict the oscillation amplitude. The source of this difference may be deformations of the rotor blades and tower in these conditions, which are not captured by numerical models, as they assume a rigid geometry. Such differences are more pronounced for higher oscillations amplitudes and consequently larger accelerations during the cycle of motion, which could indeed amplify deformations of the turbine model. In terms of phase shift, no clear trend is observed in the experimental results as the phase shift oscillates around the -90° quasi-steady

value. Possible differences fall within the uncertainty of the phase calculation, which was evaluated as about 5°. The numerical results are in agreement with the experimental data, as the differences in phase shift across the operating conditions are within the 5° range.





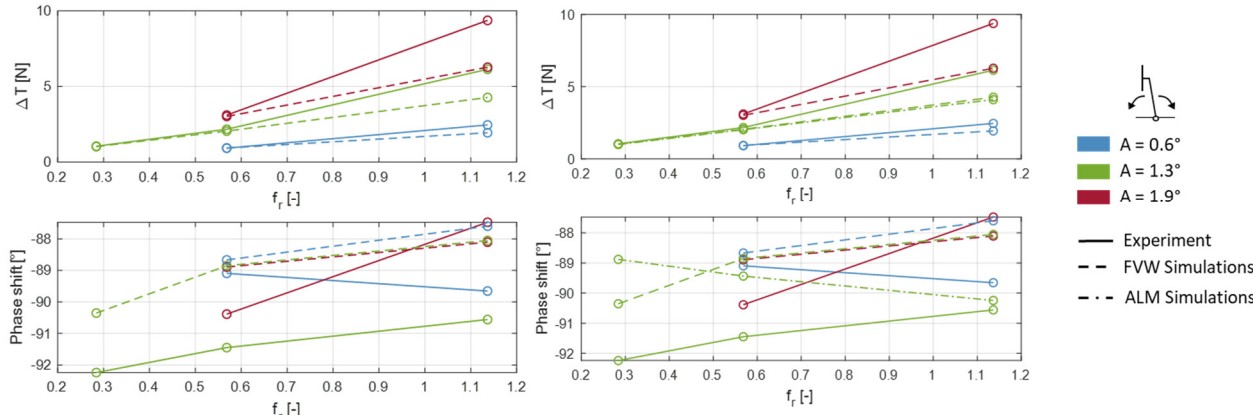

**Figure 7 Amplitude and phase shift of thrust and torque oscillations for surge motion cases.**

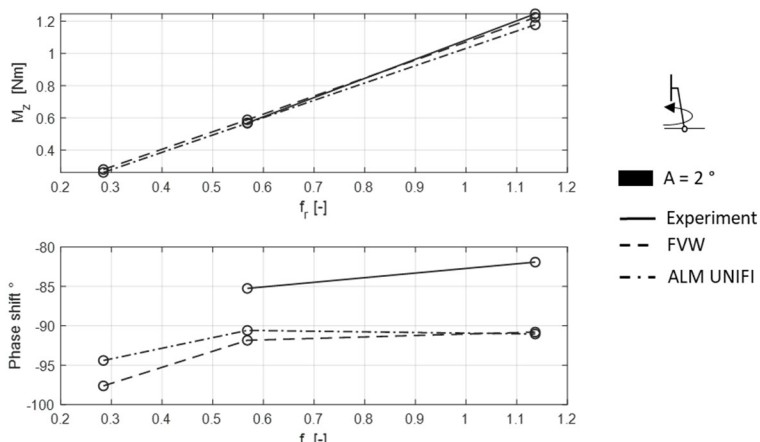


**Figure 8 Amplitude and phase shift of thrust and torque oscillations for yaw motion cases.**

Different frequencies of platform motion were investigated for the yaw motion cases, with a constant amplitude of 2°. The yaw motion causes periodic oscillations of the momentum around the z axis at the frequency of platform motion. The FVW and ALM method by UNIFI correctly capture these oscillations in all the investigated frequencies of motion (Figure 9).

Finally, the wake response was evaluated for different misalignment angles between platform motion and wind direction. In these cases, the rotor thrust oscillates at the frequency of platform motion proportionally to the apparent velocity in the wind direction. For this reason, the amplitude of thrust oscillations reduces with the misalignment angle. The numerical methods, not shown for brevity, show only minor differences with the experimental data with increasing misalignment angle.





## 3.2 Spanwise results with blade-resolved CFD

The capability of the low-fidelity models to correctly predict rotor aerodynamic performance was further investigated by comparing the FVW and ALM results with the available blade-resolved data. In fact, the blade-resolved simulation provides additional spanwise information, complementing the available data from the experiment. The additional assessment was performed for a pitch motion of 0.3° and a frequency of 2Hz.

The mean normal and tangential spanwise load distribution across the numerical methods was compared.

In terms of normal loads, the FVW and ALM simulations show good agreement with the blade-resolved simulation over most of the blade span, as these methods correctly capture the mean normal load even close to the root and tip regions. Some differences are observed in the central part of the blade, where the FVW and ALM simulations similarly overestimate the normal loads. Additionally, some differences are observed between the two ALM methods employed. Such difference might be due to the different sampling of the angle of attack, especially in the root and tip regions, where 3D effects become

predominant and these methods do not account for the contribution of chordwise bound circulation (Rahimi et al., 2018). At the tip, further differences are also probably induced by the different approaches of the two ALM model, as POLIMI does not include a tip losses model. Finally, differences in the insertion of the forces in the CFD domain might further increase the differences across the two models. In terms of tangential loads further differences are observed across the methods. The ALM by UNIFI overestimates the tangential force over most of the blade span while the FVW and ALM implementation by POLIMI

show better agreement with the blade-resolved simulations. Nevertheless, the most significant differences across the numerical results are observed in the first 30% of the span. Both FVW and ALM underestimate the tangential loads in this region significantly. This difference is probably caused by a combination of three-dimensional effects, which are not captured in the lifting line models, and inaccuracy in the aerodynamic polars. In fact, in the root region, the blade profile is obtained an interpolation of the SD7032 profile and a cylindrical section and the polars are consequently computed by interpolation,

possibly leading to significant inaccuracies. In fact, despite the differences int the tangential loads, the mean angle of attack calculated from the ALM in this region is in better agreement with the blade-resolved data, in comparison to the FVW simulations (Figure 11). The angle of attack distribution (Figure 11) also shows how the FVW and ALM simulations can correctly predict the angle of attack in the inner part of the blade, where three-dimensional effects are negligible. Despite the differences in the aerodynamic loads, the two algorithms for the definition of the angle of attack, namely the line average and

vortex methods, provide almost identical results in terms of mean angle of attack distribution, which would suggest that the differences observed between the ALM simulations are mostly driven by the different insertion of the aerodynamic loads in the CFD domain.

The periodic oscillations in aerodynamic loads, which were investigated in Sect. 3.1, are not homogenous along the span. For this reason, the amplitude and phase shift of the normal force oscillations along the span was evaluated using the Fourier

transform. Results showed that the maximum of the oscillations amplitude is located at about 70% of the span for the blade resolved simulations. Instead, the low fidelity models show the maximum between 50% and 60% of the span. This difference




could be connected to the inaccuracies in the airfoil polars, altering the aerodynamic response to variations in the angle of attack. Additionally, differences are observed between the three rotor blades, which are caused by the effect or rotor tilt on the angle of attack, as described by Cioni et al. (2025). The FVW and ALM show an analogous difference across the blades,

showing that these methodologies can correctly predict the impact of the rotor pitch on the different blades. In particular, results show that the FVW approach, if properly tuned, can lead to similar results to the ALM approach in terms of rotor loads, at a fraction of the cost. The ALM are also affected by the tuning parameters and model employed, despite using the same airfoil polars. Indeed, the ALM implementation by UNIFI shows better agreement with the blade-resolved simulation in the first 40% of the span, while POLIMI shows improved results between 60% and 80% of the blade, even though such difference

is observed only for one of the rotor blades.

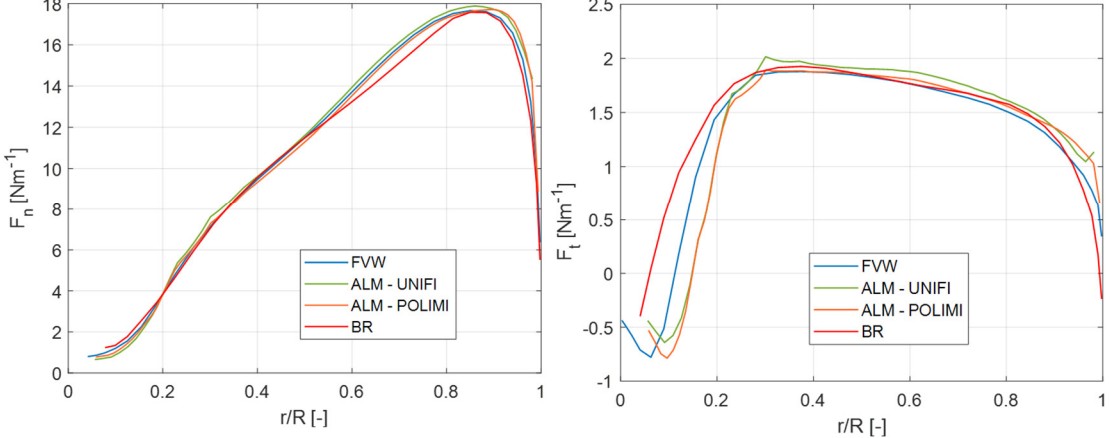

**Figure 9 Spanwise distribution of normal and tangential loads averaged over a cycle of platform motion.**

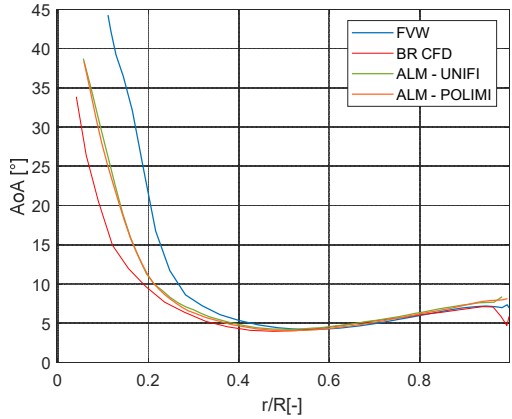

**Figure 10 Mean spanwise distribution of the angle of attack for blade 1.**





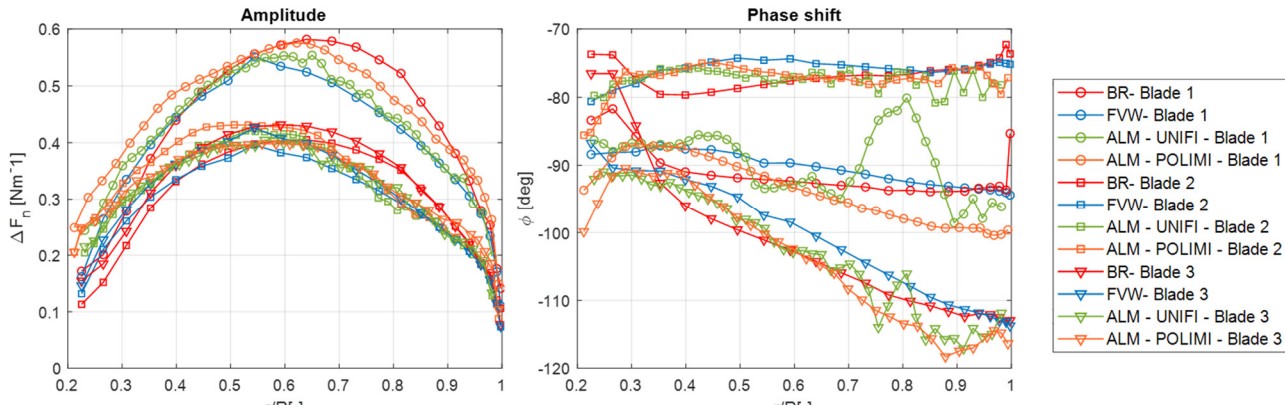


**Figure 11 Distribution of amplitudes and phase shifts of the normal loads along the span.**

## 4 Multi-fidelity analysis of wake dynamics

The impact of platform motion on the wake response was evaluated for all the test cases described in Sect. 2. First, the mean wake response was evaluated by averaging the results over multiple cycles of platform motion (Sect. 4.1). The mean wake

deficit and turbulent intensity profiles from the experiment and numerical methods were compared. Then, the onset of velocity oscillations was evaluated, to gain further understanding on the wake dynamics (Sect. 4.2). Finally, the vortex structures in the wake were investigated, to improve the current understanding on wake dynamics under sinusoidal platform motion conditions (Sect. 4.3).

### 4.1 Mean wake response

The mean wake response of the floating wind turbine model under the different platform motion conditions was evaluated from experimental results and numerical simulations at a distance of 3D from the rotor for a horizontal traverse located at hub-height (see Figure 2). For a pitch motion of the turbine, the experimental results show that platform motion has only a minor impact on the average streamwise profile (Figure 12). Analogous results were observed for the remaining operating conditions investigated in this work, which are not shown here for brevity. This result confirms the previous analysis of the OC6 phase

III project, suggesting that the sinusoidal surge and pitch motions do not lead to a significant improvement in wake recovery. Both FVW and ALM CFD simulations show an analogous wake response, as only minor differences are observed across the investigated conditions and the fixed-bottom results. The main discrepancies are observed for the FVW results which show improved mixing at the center of the wake for a reduced frequency of 0.6. In these operating conditions the FVW method might overestimate the impact of platform motion on the wake.

The differences between simulations and experiment are connected to the limitations of the numerical methods in capturing the fixed-bottom wake recovery and not differences induced by platform motion. Both FVW and CFD ALM results underpredict the wake transition at a distance of 3D, showing a bi-gaussian velocity profile ( Figure 12). In contrast, the wind



tunnel tests show improved mixing at the center of the wake and an asymmetric velocity profile, which might be due to the interaction between the rotor wake and the wake of the nacelle and tower (Fontanella et al., 2024).

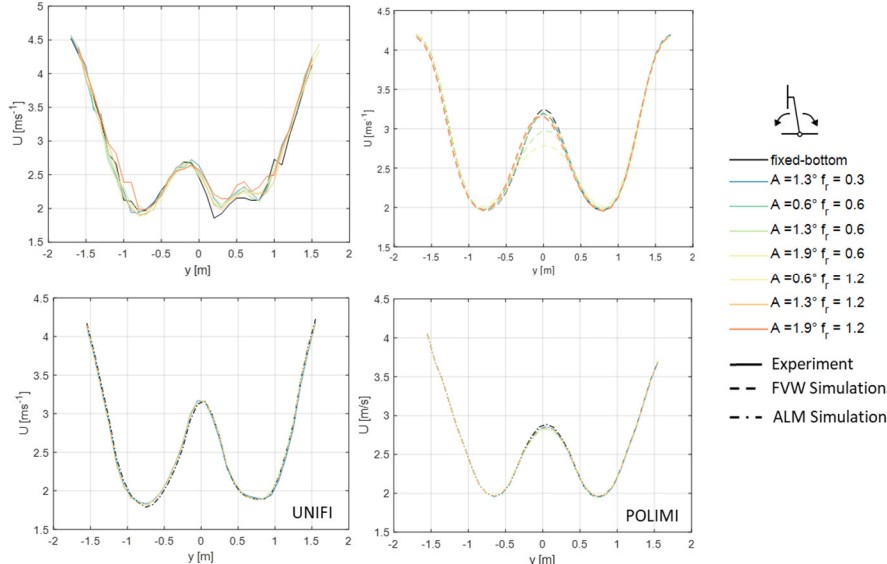


**Figure 12 Horizontal profiles of streamwise velocity at a distance of 3D from the rotor for a pitch motion of the rotor.**

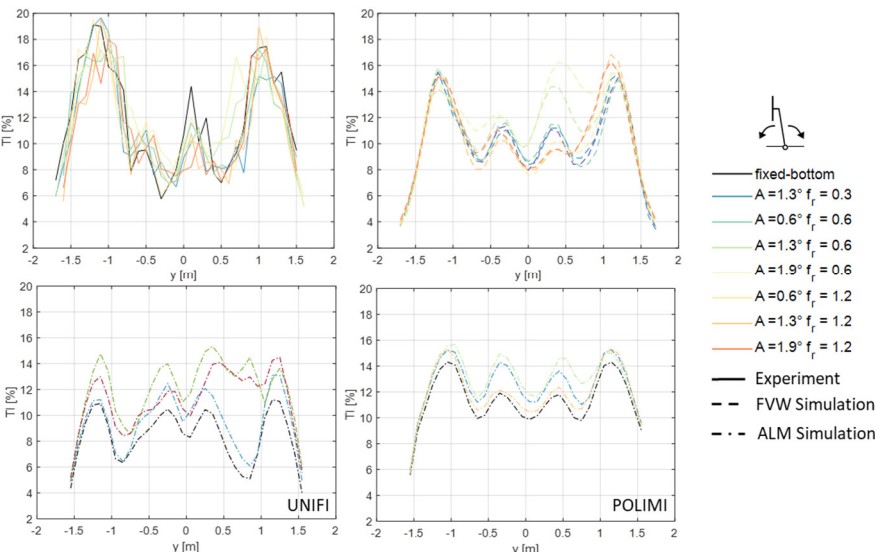

**Figure 13 Turbulent intensity horizontal profile at 3D from the rotor for the surge motion cases.**

The motion of the turbine could also lead to increased turbulence intensity in the wake, due to the improved mixing and
additional impact of periodic apparent velocity oscillations. In pitch motion conditions, an increase in turbulent intensity is observed mainly in the inner part of the wake (Figure 13). Most differences from the fixed-bottom case are observed at high reduced frequency and high amplitudes of motion (i.e. for large apparent velocity oscillations). The numerical methods are





able to capture the turbulence intensity increase observed in experiments, with minor differences. Both the FVW and CFD ALM methods show a significant increase also at a reduced frequency of 0.6 and an amplitude of 0.048m, overestimating the
impact of these conditions in comparison to the experiment. The CFD ALM results significantly underestimate the turbulence intensity in fixed-bottom conditions, due to the URANS methodology employed, in agreement with the existing literature. For this reason, this method over-predicts the impact of platform motion on the results in terms of turbulence intensity both in the center of the wake and in the shear layer, if results are compared with the ones obtained for fixed-bottom conditions.

For a sinusoidal pitch motion of the platform the increase in turbulence intensity in the wake is more significant for two specific
operating conditions: a reduced frequency of 0.6 and an amplitude of motion of 1.9° and at a reduced frequency of 1.2 and an amplitude of 0.6°. Hence, the increase in turbulence intensity is not strictly correlated with an increase in reduced frequency or in the amplitude of the apparent velocity oscillation. Instead, the wake shows an amplified response for a reduced frequency of 0.6. The FVW and ALM simulations are able to capture this trend, even if the increase in turbulence intensity is overestimated in comparison to the experiment. Some differences are across the two ALM methodologies compared in Figure
14. First of all, POLIMI shows increased turbulence intensity corresponding to the tip vortices. Such difference might be connected to the Sorensen tip losses model, which is employed in the UNIFI results but not in the ones obtained by POLIMI. Additionally, differences in the spreading of the aerodynamic forces may alter the size and strength of these vortex structures affecting the distribution of the turbulent intensity. Additionally, the ALM results show different wake responses to a pitch motion of the platform. Indeed, UNIFI shows an a-symmetrical increase in turbulence intensity, matching the experiment,
while a uniform increase is observed for the POLIMI simulations.

The wake response of the model wind turbine was evaluated also at a distance of 5D, to evaluate the impact of platform motion further downstream. In fact, the perturbations introduced in the wake due to platform motion may be amplified as the wake is convected downstream, leading to further differences between the fixed-bottom and platform motion cases. Figure 14 shows the mean streamwise velocity profile under different frequencies of pitching motion. Both experiments and FVW simulations
show only minor impact of pitching motion on the average velocity profile and wake deficit. The wake profile from the FVW and ALM simulations is not fully transitioned and shows a bi-gaussian profile. In contrast, the experiment shows further transition of the wake. Nevertheless, both experimental and FVW and ALM simulations by UNIFI show a non-symmetric wake response when the reduced frequency is 0.6, suggesting that in such operating conditions the wake dynamics are more influenced by platform motion, despite minor impacts on the overall wake deficit. This is confirmed by the turbulence intensity
profile, as the most significant increases are observed for this motion frequency (Figure 15). The ALM simulation by POLIMI show improved transition of the wake and better agreement with the experiment in terms of fixed-bottom wake deficit; however, the impact of platform motion is underestimated with regard to the experiment.

In terms of turbulence intensity, significant differences are observed across the ALM simulation. As both models underestimate the TI values in the wake due to the URANS methodology employed, the ALM by UNIFI shows significant increases in
turbulence intensity when a pitching motion of the platform. Such difference with the fixed-bottom response is significantly reduced in the simulations by POLIMI. To investigate the differences between the two ALM models, UNIFI run some tests





without including the nacelle, tower and robot; however, no significant difference was observed in the result. Additionally, the different kernel size between POLIMI and UNIFI is probably not the root cause of the observed differences, as additional tests run by UNIFI  matching the setup by POLIMI do not show significant differences. Another possible reason could be the

different meshing strategies employed by UNIFI and POLIMI but further tests are required to confirm the root causes of these differences.

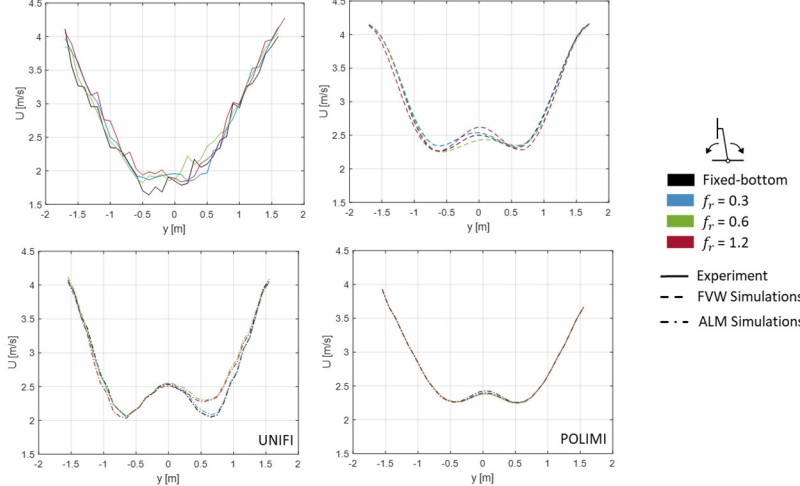

**Figure 14 Mean horizontal streamwise velocity profile at 5D from the rotor.**

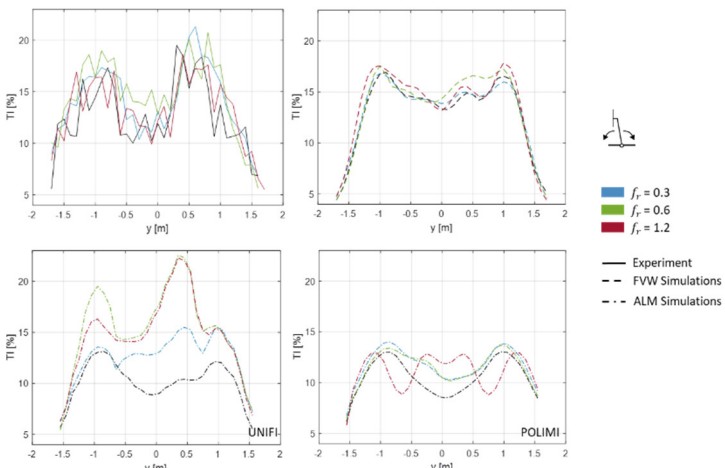

**Figure 15 Horizontal turbulence intensity profile at a distance of 5D from the rotor under pitching motion of the turbine.**

During the NETTUNO wind tunnel campaign, the initial analysis of surge and pitch cases was expanded by including different types of platform motion. The wake response was evaluated also under sinusoidal yaw oscillations at a constant amplitude and with varying frequency. In fact, surge and pitching motions of the platform mainly cause periodic oscillations in the apparent wind speed observed by the rotor. However, yaw motion may also lead to wake meandering and improved mixing.



Nevertheless, similarly to the surge and pitch motions, experiments and numerical simulations show that yaw motion does not
lead to significant improvements in wake deficit.

In terms of turbulence intensity (Figure 16), yaw motion enhances turbulent mixing in the inner part of the wake, similarly to
what was observed for surge and pitch motions. The most significant increase in the experimental results is observed for a
reduced frequency of 0.6, while almost no difference is observed at a higher frequency of motion and minor increases are
observed at $f_r = 0.2$. The FVW and ALM simulations generally overpredict the impact of platform motion across the wake,
even though the reduced frequency of 0.6 shows the most significant turbulence increase in the wake, as the turbulence intensity
maxima corresponding to the wake shear layer become wider, suggesting an increase in the boundary layer thickness.
Significant increases are observed in all the numerical methods also for a reduced frequency of 0.2, which are not shown in
the experimental data, suggesting that the wake dynamics are not captured in these conditions.

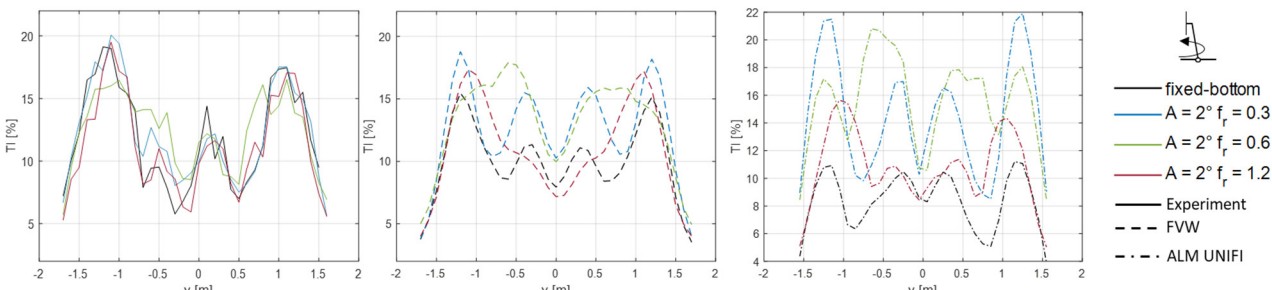


**Figure 16 Turbulent intensity profiles at a distance of 3D during yaw motion of the wind turbine.**

Finally, the wake response of the model wind turbine was investigated in more complex platform motion conditions, where
the wind and wave are misaligned, resulting in a combination of surge and sway. In these test cases the mean wake response
was evaluated as a function of the misalignment angle under fixed amplitude and frequency of motion (Figure 19). A reduced
frequency of 0.6 was chosen as this resulted in the most significant differences in wake response in comparison to the fixed-
bottom case.

Even under a combination of the surge and sway the impact of platform motion on the average wake deficit is limited, despite
the introduction of the crosswise motion. This result is confirmed across experiments and simulations with good agreement.
However, an increase in turbulence intensity is observed in comparison to the fixed-bottom case. For a combination of surge
and sway the turbulence intensity is increased mostly in the inner part of the wake. In the experimental data, the turbulence
intensity increases with the misalignment angle, up to 90° (i.e., pure sway), even though the differences are limited when
increasing the misalignment angle from 30° to 45° and 90°. The FVW and ALM simulations show a similar increase in the
turbulence intensity in the inner part of the wake; however, the increase in turbulence intensity is overestimated, especially
when a pure sway motion is considered. The ALM results show further differences in the shear layer in comparison to the
experiment as the turbulent intensity in the shear layer increases significantly due to platform motion.



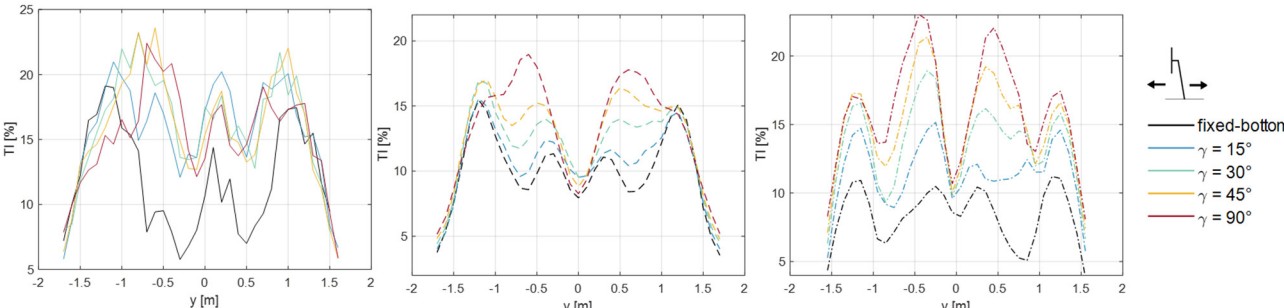

**Figure 17 Turbulent intensity horizontal profiles for wind-wave misalignment cases at 3D from the rotor. Results are shown for varying angles of misalignment.**

The analysis of the mean wake response of the wind turbine model under different types of platform motion conditions suggests
that platform motion only has a limited impact on wake recovery up to a distance of 5D from the rotor. The additional mixing in the wake induced by the motion leads to increased turbulent intensity in the wake, which could be detrimental to downstream machines, but not to improved wind speed recovery. Numerical methods correctly predict the average wake dynamics but may overpredict the increase in turbulence intensity, both for FVW and ALM simulations. In this case, the higher fidelity method does not show significant improvements in the agreement with the experiment, possibly due to the limitations of the URANS
approach and ALM setup employed in this work.

## 4.2 Unsteady wake response

Platform motion may affect the wake response of the wind turbine by inducing periodic velocity oscillations or deflections in the wake. Such oscillations may not impact on the average wake response but may be detrimental for downstream turbines. In this section the unsteady wake response of the turbine is investigated.
The periodic oscillation of the turbine induces oscillations in the relative velocity seen by the rotor and consequently periodic oscillations in rotor thrust and wake deficit in the wake. For example, Figure 21 shows the phase-averaged wake deficit calculated along a horizontal traverse at 3D from the rotor during a cycle of surge motion at a reduced frequency of 0.6 and an amplitude of 0.032m. A clear sinusoidal trend at the frequency of platform motion is observed both in the experiment and the numerical results. Hence, to quantify the entity of such oscillations, for each of the operating conditions investigated in this
work, the amplitude of the wake deficit oscillations was calculated using the Fourier transform to extract the contribution at the frequency of motion.



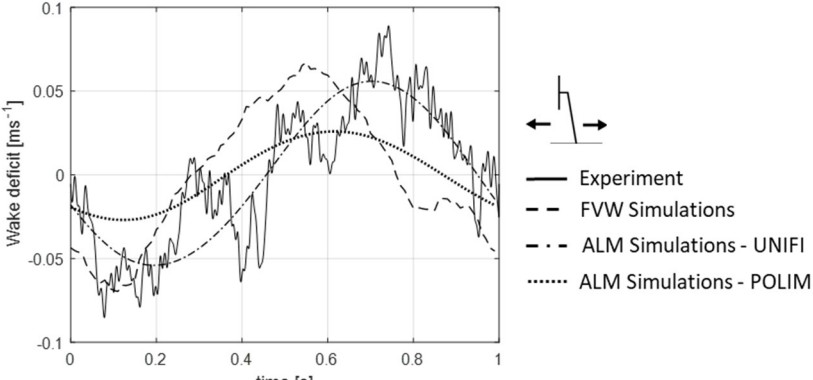

**Figure 18 Phase-averaged wake deficit time series during a cycle of surge motion at $A = 0.032\ m$ and $f_r = 0.6$.**

Figure 19 shows the amplitude of the wake deficit oscillations along the horizontal traverse at 3D from the rotor, for the surge
and pitch cases. In the case of surge motion, the experimental results show a maximum of the wake deficit oscillations at the
reduced frequency of 0.6 at the largest amplitudes of platform motion. This result indicates that the velocity oscillations in the
wake are not proportional to the rotor thrust oscillations induced by the motion, which increase almost linearly with the
frequency of oscillation (see Sect. 3.1). Such a similar linear trend is observed in terms of wake deficit oscillations only for an
amplitude of 0.016m.

At large amplitudes of platform motion (i.e., 0.032 m and 0.048m) the numerical results show only minor differences with the
experimental data up to a reduced frequency of 0.6. At high reduced frequency, the FVW simulations show an almost linear
increase in the wake deficit oscillations, in contrast to the reduction observed in the experimental tests. In the ALM data, the
wake deficit oscillations do not show a similar increase but still overestimate the velocity oscillations recorded in the
experiment. As observed in Sect. 4.1, platform motion at the reduced frequency of 0.6 excites the wake more significantly than
in other conditions and the velocity oscillations in the wake increase with reduced frequency only when the apparent velocity
oscillations are low, at about 0.2 ms$^{-1}$.

For the pitch cases, a reduction of the wake deficit oscillations at high reduced frequencies is observed for all the amplitudes
of motion, independently of the apparent velocity value. The FVW simulations show an almost linear increase of the velocity
oscillations with reduced frequency; however, for a pitch motion the amplitude of the wake deficit is underestimated rather
than overestimated in comparison to the surge cases. At high reduced frequency a further increase of the velocity oscillations
is observed, suggesting that the FVW method might be unreliable in predicting the wake dynamics in these conditions.
Similarly to the surge cases, the ALM simulations show a closer response to the experiment to variations in the reduced
frequency, as the velocity deficit oscillations are maximized for a reduced frequency of 0.6. However, significant differences
are observed across the two ALM methodologies employed in this work, as the POLIMI simulations usually underestimate
the onset of the velocity oscillations in the wake in comparison to the experiment and to the ALM simulations by UNIFI.




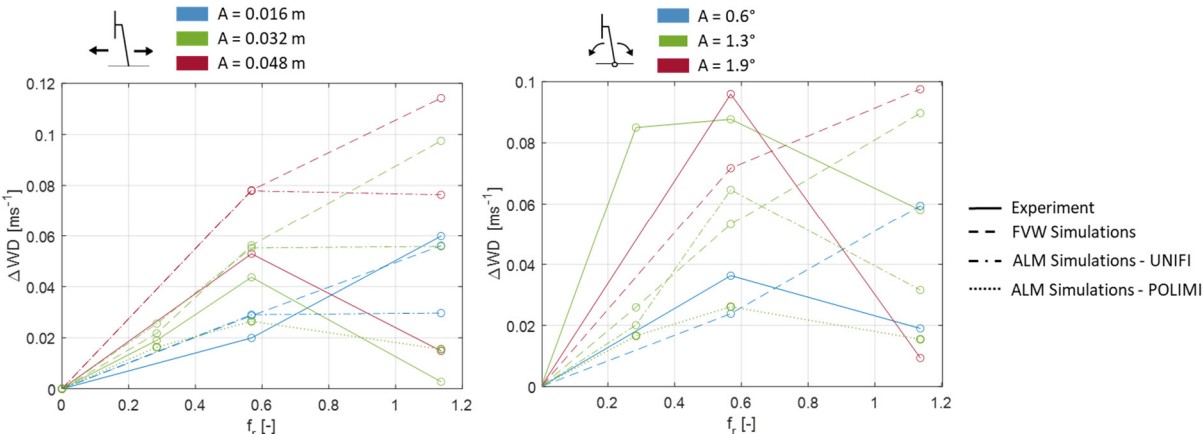

**Figure 19 Wake deficit oscillation amplitude for surge and pitch motions.**

Since the wake deficit only shows an integral evaluation of the velocity oscillations in the wake, an in-depth analysis of the spatial distribution of the velocity oscillations was performed. Figure 20 shows the velocity oscillations along the horizontal
traverse during a cycle of surge motion. For both experiments and numerical simulations, the velocity oscillations are not uniform across the wake. At low reduced frequency almost no oscillation is observed, as confirmed by both experiments and simulations. When the reduced frequency is increased to 0.6, the whole wake pulsates at the frequency of platform motion, and the amplitude of oscillations is also increased. However, when the reduced frequency is 1.2, significant velocity oscillations are observed in the experimental results only in the outer part of the wake. The CFD ALM simulations show a similar
distribution of velocity oscillations, while the FVW method still shows significant oscillations in the inner part of the wake. These results show how the wake dynamics are altered with increasing reduced frequency, resulting in different distribution and intensity of the velocity oscillations in the wake.

When a pitch motion of the platform is considered, the entity of the velocity oscillations in the wake is increased (Figure 19). Such an increase might be connected to the periodic deflections of the wake due to the pitch motion. In the experimental
results, the motion induces a periodic pulsation which is more significant at the reduced frequency of 0.6, analogously to the surge motion case. However, in this case the velocity oscillations are not symmetric around the center of the wake but are more significant on the right-hand side. This result is confirmed in both FVW and ALM simulations and might be connected to the combination of rotor rotation and pitch motion, leading to an asymmetrical response. At a reduced frequency of 1.2, the velocity oscillations are significant only at the edges of the wake in the experimental results; however, the FVW simulations show
significant oscillations also in the inner part of the wake, similarly to the surge motion cases. Across the ALM simulations, UNIFI and POLIMI show similar spatial distribution of the velocity oscillations of the wake, which match the experiment, but the entity of such pulsations is affected by the methodology employed. In general, the ALM by POLIMI underestimates the onset of velocity oscillations in the wake in comparison to the simulations performed by UNIFI.

No significant differences were observed in the velocity oscillations along the vertical transverse at 3D both for surge and
pitch cases, showing that the velocity oscillations do not have a preferential direction.



**Figure 20 Wake velocity oscillations along a horizontal traverse at 3D from the rotor during a cycle of surge and pitch motion.**

As the wake is convected downstream, also the velocity oscillations are propagated. The velocity oscillations in the wake could be amplified as they are convected downstream. For this reason, the wake response under pitch motion was investigated also
at a distance of 5D from the rotor.



The distribution of the velocity oscillations during a cycle of pitch motion provides further insight into the wake dynamics at 5D from the rotor (Figure 24). At low reduced frequencies ($f_r = 0.3$), both experiment and simulations show an amplification of the velocity oscillations with the distance from the rotor, as the right side of the wake shows a coherent pulsation of the wake at the frequency of platform motion.

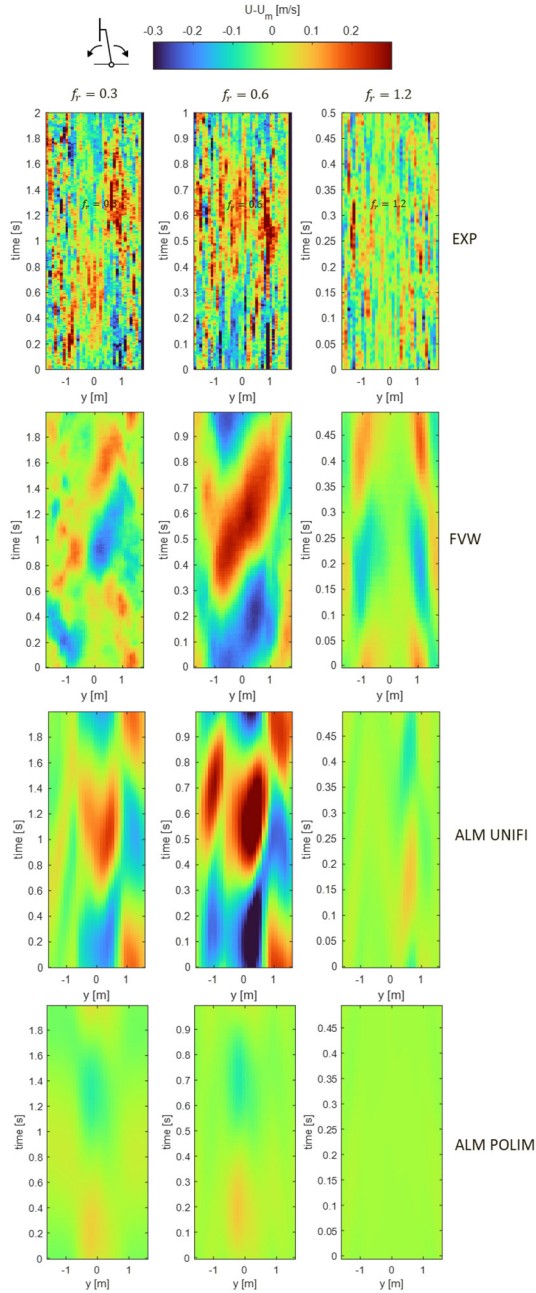


**Figure 21 Wake velocity oscillations along a horizontal traverse at 5D from the rotor during a cycle of pitch motion.**

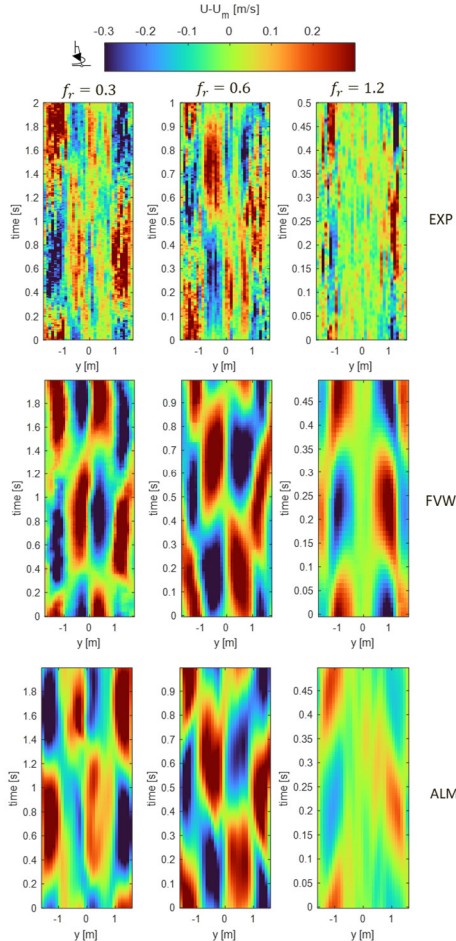

**Figure 22 Wake velocity oscillations along a horizontal traverse at 3D from the rotor during a cycle of yaw motion.**

At high reduced frequency ($f_r = 1.2$) a substantial dissipation of the velocity oscillations is observed, as these are mainly

present in the shear layer of the wake. However, at a reduced frequency of 0.6, the wake shows a synchronization to the motion

frequency, as both experiments and simulations show an almost homogenous pulsation of the wake. However, the FVW and

ALM UNIFI results overestimate such oscillations, while the ALM by POLIMI underestimates them, suggesting that the

employed methodologies may struggle to capture the propagation of the velocity oscillations as they are convected

downstream.

The onset of velocity oscillations was also investigated for a yaw motion of the platform. The wake response as a function of

platform motion frequency is similar to the one observed under surge and pitch motions, as the velocity oscillations are

predominant at a reduced frequency of 0.6 and involve the whole wake, while at a reduced frequency of 1.2 only the shear

layer shows significant velocity oscillations (Figure 21) . In addition, yaw motion affects the wake also at a reduced frequency

of 0.3, showing significant velocity oscillations, mainly in the shear layer. The FVW model correctly captures the spatial





distribution of the velocity oscillations, however the intensity of the wake response is overestimated in comparison to the
experiment. The ALM shows better agreement with the experiment, especially at low and high reduced frequencies.

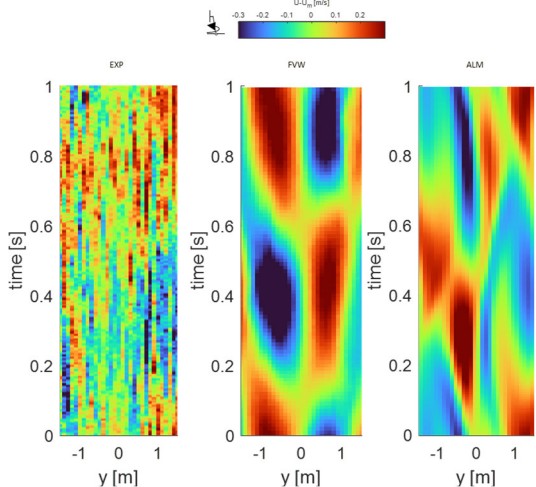

**Figure 23 Wake velocity oscillations along a horizontal traverse at 5D from the rotor during a cycle of yaw motion with an amplitude**
**of 2°.**

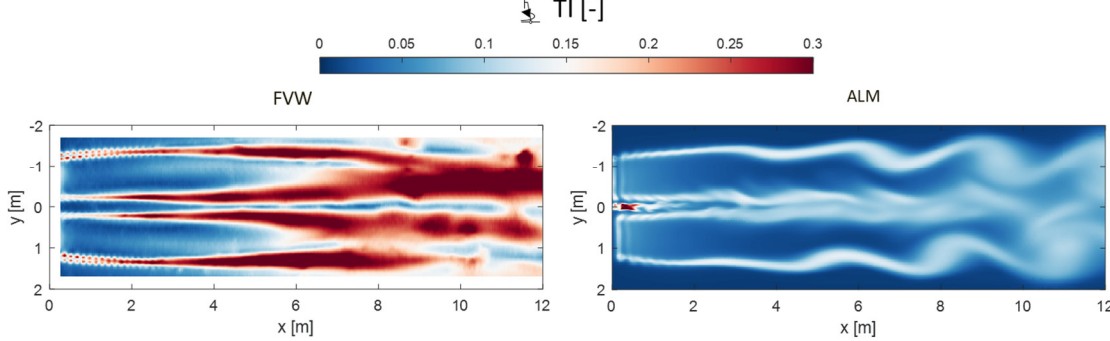

**Figure 24 Turbulent intensity distribution under sinusoidal yaw motion of the rotor at a reduced frequency of 0.6.**

The velocity oscillations observed for a sinusoidal yaw motion of the turbine are caused by meandering of the wake, introduced
by the periodic yaw oscillations of the wind turbine model. In fact, the main difference between surge (or pitch) motions and
yaw oscillations concerns the different dynamics of the wake. For the former, the pitch and surge motion introduce periodic
oscillations in rotor thrust which consequently induce a pulsation of the wake; for the latter, the motion introduces periodic
deflections of the wake in the side-to-side directions due to the yaw misalignment of the rotor. For this reason, the velocity
oscillations under yaw motion are anti-symmetrical with respect to the wake center, as the wake is deflected from left to right.
Such differences can have a significantly different impact on downstream turbines. In fact, for a surge and pitch motion the
whole wake pulsates at the same frequency and phase, which could lead to significant oscillations in the rotor thrust and power.




In contrast, for a yaw motion the anti-symmetrical velocity oscillations induced by wake meandering might compensate over the whole rotor, resulting in smaller power oscillations. Nevertheless, significant load oscillations might be induced on both sides of the rotor, possibly increasing fatigue loading.

The wake response under yaw motion was also investigated at a distance of 5D from the rotor to understand how the deflection of the wake is propagated downstream. In the experiment the velocity oscillations are not dissipated with increasing distance from the rotor. However, the oscillations are no longer anti-symmetrical with respect to the centre of the wake, in contrast to what was observed at 3D. This shows how, as the wake is convected downstream, it shows a significant pulsation at the frequency of motion. The FVW simulations do not show a similar wake dynamics and the velocity oscillations are anti-symmetrical, similarly to what was observed at a distance of 3D. The limitation of this approach might lead to inaccurate prediction of rotor loading on downstream machines. The ALM show a closer distribution of the velocity oscillations to the experiment; however, a fully uniform pulsation of the wake is not observed. In this case the ALM method may overpredict the distance at which the wake pulsates at the frequency of motion.

The different wake dynamics between the FVW and ALM simulations can be observed from the distribution of the turbulent intensity in the wake (Figure 24). Until a distance of about 1D the numerical methods show a similar wake response. However, as the vortex structures in the wake breakdown the FVW method shows a significantly different turbulent intensity distribution than the ALM simulation. In the latter, the wake shear layer pulsates at the frequency of platform motion. Such oscillations are amplified until they lead to a uniform oscillation of the whole wake at about 5D.

In the cases of wind-wave misalignment, platform motion causes an additional side to side displacement of the turbine which could lead to significant oscillations in the wake.

The introduction of the wind-wave misalignment leads to an increase of the velocity oscillations in the wake in comparison to the aligned surge cases (Figure 25). In addition, velocity oscillations are anti-symmetric with respect to the wake center, suggesting that the main driver of velocity oscillations is wake meandering (i.e., side to side motion of the wake) rather than pulsation of the wake due to periodic oscillations in rotor thrust. In these operating conditions further differences are observed between the experiment and simulations, in comparison to the surge cases. In particular, the velocity oscillations in the wake obtained from FVW and ALM simulations do not show a similar pattern to the experiment. This result might indicate that numerical methods may struggle to capture a side-by-side motion of the wake caused by a significant misalignment angle between the wind and platform motion direction.





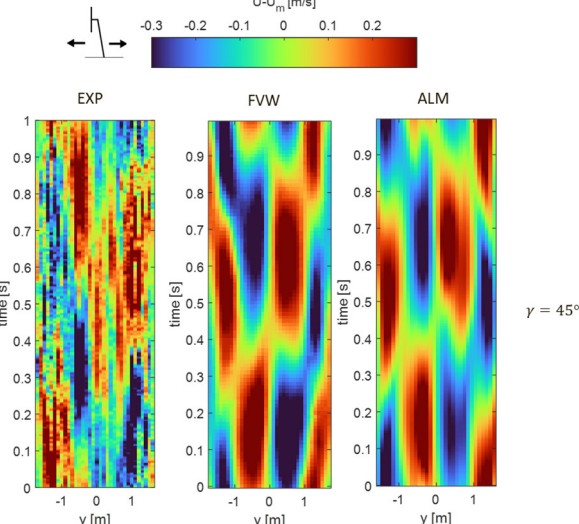

**Figure 25 Wake velocity oscillations along a horizontal traverse at 3D from the rotor during a cycle of surge motion at a constant reduced frequency of 0.6.**

More differences between the numerical methods are observed at a distance of 5D from the rotor. In this case, the experiment shows distinct velocity oscillations induced by the motion, but their intensity is reduced in comparison to those recorded at 3D. The FVW simulation shows instead a significant intensity of the wake velocity oscillations and only the ALM shows a similar reduction to the experiment. This result suggests that the FVW method can capture the onset of the velocity oscillations in these conditions only close to the rotor, but the dissipation of these oscillations is better predicted by high-fidelity models, such as ALM CFD.

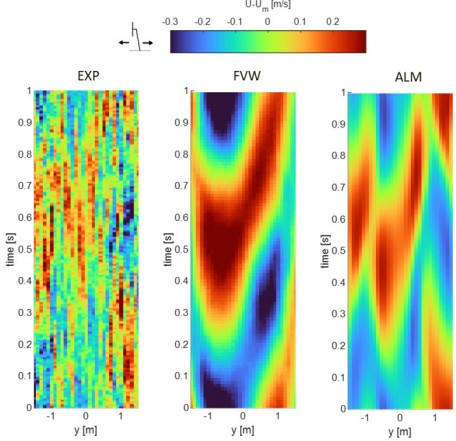

**Figure 26 Wake velocity oscillations along a horizontal traverse at 5D from the rotor during a cycle of surge motion at a constant misalignment angle of 30°.**





## 4.3 Wake vortex structures

In Sect. 4.2, it was observed that the wake shows a different dynamic response with varying reduced frequency and type of motion. To improve the current understanding on wake dynamics of a floating wind turbine the vortex structured in the wake are investigated levereging the additional data provided by the FVW and ALM simulations. In fact, the numerical methods allow the analysis of a full 2D slice of the wake, which would require complex measurement technicques in the experiment, such as PIV.

The analysis of 2D velocity fields can be used to investigate the formation of coherent vortical structures, which are responsible for the breakdown and recovery in the wake. Multiple methods have been proposed in the literature to investigate vortex structures; however, the Q-criterion, is used in this work, as it allows the distinction between vortex structures and wake shear layers.

Figure 27 show the vortex structures in the wake of the wind turbine model under imposed surge motion and for varying reduced frequencies, obtained from the FVW and CFD ALM simulations respectively. In the fixed-bottom configuration, the tip and root vortices are the main structures in the wake. The tip vortices form a stable helical structure which is convected downstream in the wake. Both in the FVW and CFD ALM simulation the tip vortices interact and merge after about 1D from the rotor, collapsing into complex vortex structures. As the distance from the rotor increases, the FVW simulation still shows strong vortex structures in the shear layer of the wake, which are dissipated in the CFD ALM method. In fact, the FVW method models the convection of the vortex filaments and their breakdown, while the CFD ALM solves their interaction and dissipation. This represents one of the limits of the FVW method, when investigating the far wake-behavior of a wind turbine. The FVW and CFD ALM simulations also show significant differences at the center of the wake. In fact, in the FVW simulations the root vortices form a stable helix in the inner part of the wake, and they interact and collapse only after about 1D. In contast, in the ALM simulation the root vortices are stretched and deformed very close to the rotor, possibly due to the interaction with the nacelle and tower shear layer. Such deformation leads a significantly faster breakdwon of the root vortices, whch merge togheter after about 0.5D from the rotor. This difference might be responsible of the improved transition in the ALM results in comparison to the FVW method in Sect. 4.1.

The impact of surge motion on the wake was evaluated by investigating the impact on the wake vortex structures. At low reduced frequency ($f_r = 0.3$) the imposed surge motion has a minor impact on both root and tip vortices. In fact, in Sects. 4.1 and 4.2, only minor differences were observed in the wake dynamics. At a higher reduced frequency ($f_r = 0.6$) both FVW and CFD results show a pulsation of the vortex structures at the frequency of platform motion, as the tip vortices merge togheter forming larger and less intense periodic vortex structures. Nevertheless, the tip vortices collapse at a similar distance from the rotor as the fixed-bottom case. This might explain why no siginificant difference is observed in terms of average free stream velocities in the wake. In fact, wake recovery is connected to the breakdown of the helixed formed by the tip vortices, which leads to improved mixing with the free-stream flow.





In contrast, minor differences are observed in the root vortices. In the FVW simulations no clear difference is observed in comparison to the fixed-bottom case. Instead, in the CFD results the root vortices show a small pulsation at the frequency of motion after about 3D, even though such vortex structures are weaker than those formed by the tip vortices.

At the highest reduced frequency, the tip vortices merge into large coherent structures shed at the frequency of platform motion, similarly to what was observed at $f_r = 0.6$. However, such vortex structures are smaller and more intense than those observed at lower reduced frequency. This might explain the differences observed in terms of velocity oscillations in Sect. 4.2. In fact, at $f_r = 0.6$ the larger vortex structures may cause pulsation of the wake both in the shear layer and inner part of the wake, while at $f_r = 1.2$, the smaller size of these coherent structures only leads to significant pulsation in the outer section of the

wake.

Another difference in the wake dynamics when the reduced frequency is increased from 0.6 to 1.2 concerns the dissipation of such larger vortex structures. In fact, in the CFD ALM results, at $f_r = 1.2$ these vortical structures are formed closer to the rotor but they are also dissipated faster, which is not observed in the FVW results. Such difference, which might be linked to the limitations of the FVW method to capture vortex breakdown and its interaction with free stream turbulence, might explain

the differences observed in terms of velocity oscillations. In fact, the FVW approach overpredicts the entity of velocity oscillations in comparison to the experiment and the ALM results at a distance of 3D from the rotor.

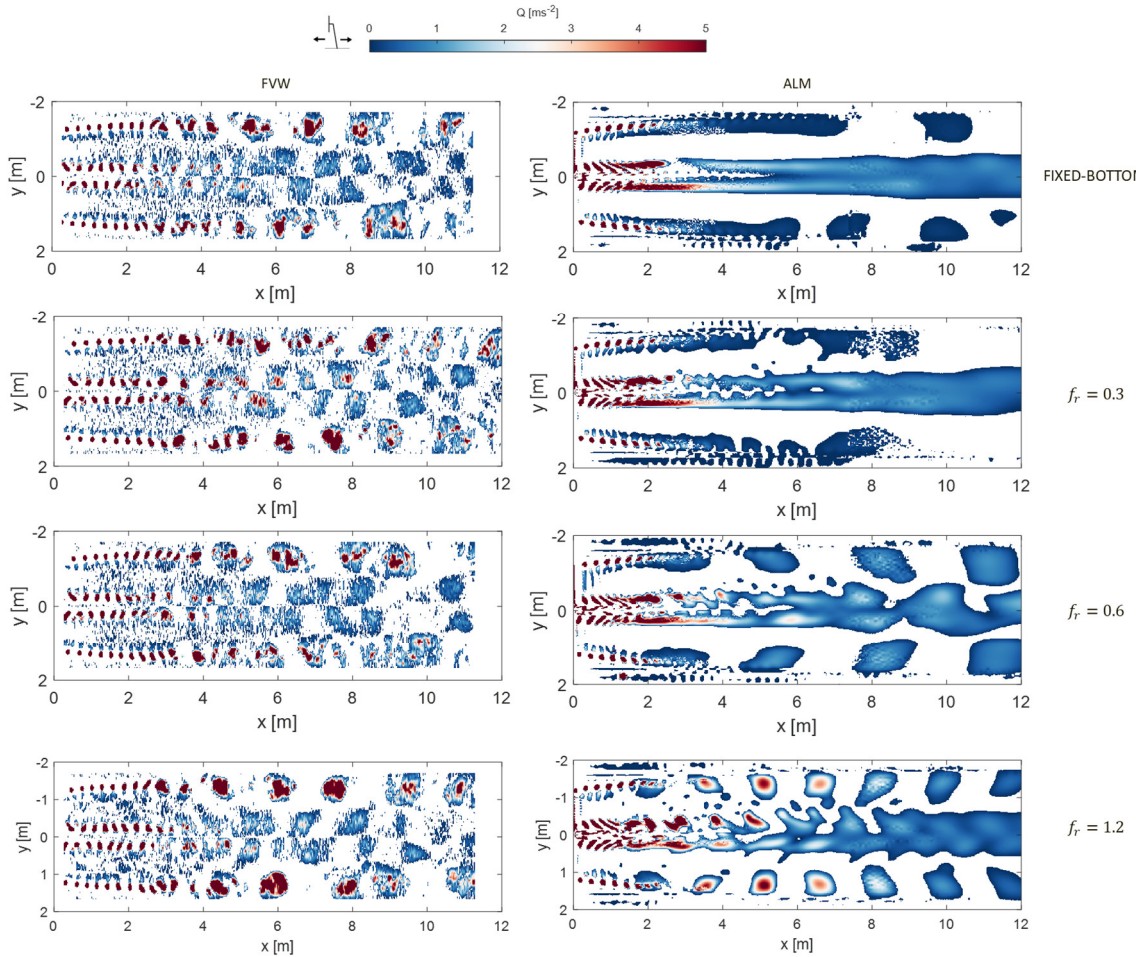

**Figure 27 Vortex structures in the wake of the wind turbine model under varying frequencies of surge motion. Results obtained from FVW and CFD ALM simulations performed by UNIFI.**

The capabilities of FVW and ALM URANS simulations to capture the vortex structures in the wake of a floating wind turbine was evaluated by comparing the results with the vortex structures observed from LES ALM simulations (Pagamonci et al., 2025). The results are shown in Figure 28 for a pitch motion of the platform. The main differences between the different numerical methodologies concern the breakdown of the wake. The LES simulation shows the breakdown of the largest structure into smaller scales, which is not shown in the URANS or FVW simulations. The large coherent structures observed

in the URANS simulations are also not easily identifiable in the LES results, as multiple smaller structures can be observed as the wake breaks down. Nevertheless, the wake starts to pulsate as the vortex strucutres formed from the tip vortices start to interact with the tip vortices and breaking down, similarly to the URANS results. This result suggest that URANS ALM simulations may predict some of the main dynamics of the wake of a floating wind turbine but might overestimate the strength of the coherence structures which are formed due to platform motion.




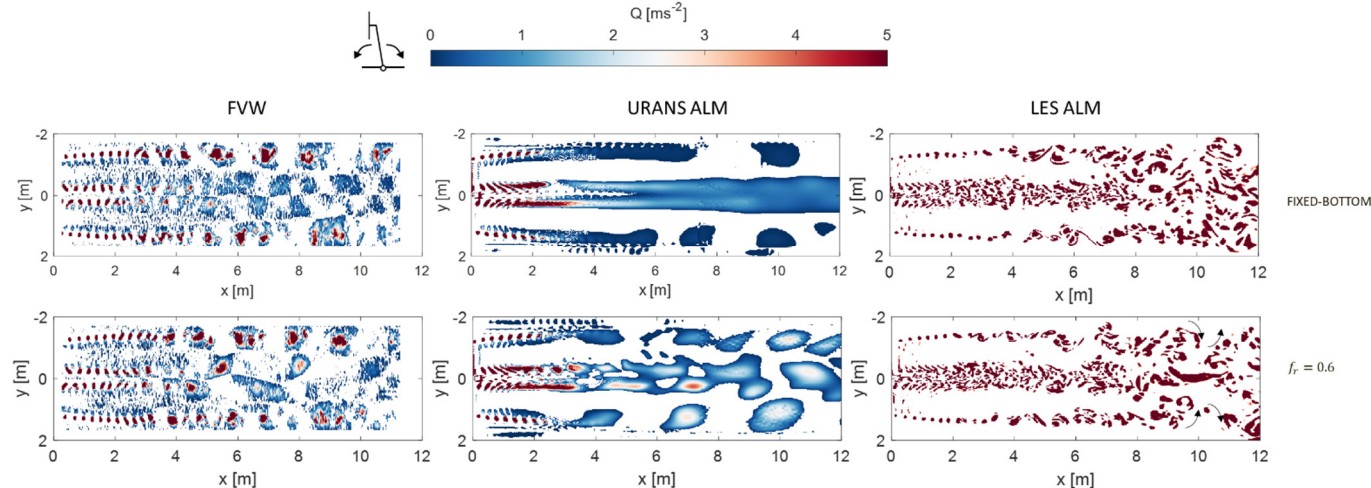

**Figure 28 Vortex structures in the wake of the model wind turbine under sinusoidal pitch motion. Arrows highlight the pulsation of the wake in the LES results.**

The impact of yaw motion on the vortex structures was also investigated, as under such platform motion conditions significant velocity oscillations were observed in Sect. 4.2. Figure 29 shows the vortex structures in the wake for the FVW and ALM simulations. Results are shown for a single platform motion frequency of 0.6, as the wake shows similar responses to changes in platform motion frequency to those observed for surge motion.

For a yaw motion of the platform, the FVW methods show the onset of significant vortex structures at the edges of the wake, due to the merging of the tip vortices. These vortex structures are alternating on the two sides of the wake, leading to an anti-symmetrical configuration. The root vortices are also merged into larger structures, even though their strength is significantly smaller. The ALM show similar results until about 2-3D from the rotor. Large coherent structures are formed both in the inner and outer part of the wake, with a similar anti-symmetrical distribution. However, differences arise with increasing distance from the rotor. The ALM results show that the coherent structures at the edge of the wake become larger as they are convected downstream, until they affect the root vortices, leading to a pulsation of the whole wake. A similar behaviour was observed from the analysis of the experimental results, showing how the ALM can better predict the wake dynamics in comparison to the FVW method. Additionally, there are significant differences between the FVW and ALM simulations in the dissipation of the above-mentioned vortex structures, which could explain the overestimation of the velocity oscillations by the former methodology observed in Sect. 4.2.



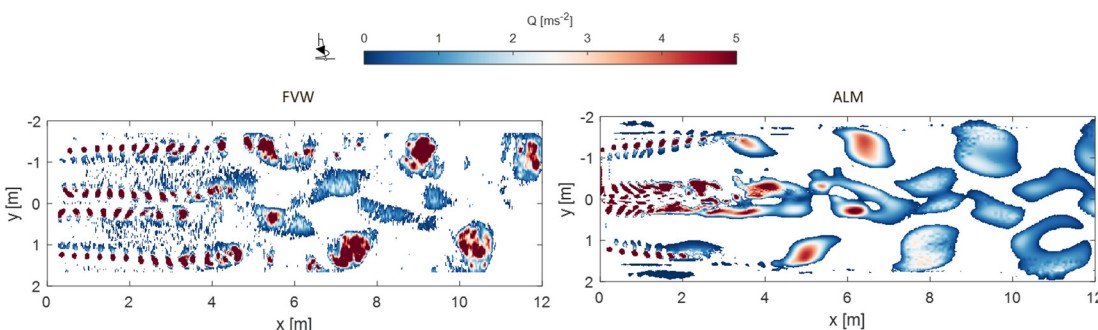

**Figure 29 Vortex structures in the wake of the wind turbine model under yaw motion. Results obtained from FVW simulations.**

Finally, the impact of misalignment angle between platform motion and the wind direction on the vortex structures in the wake was evaluated. Figure 30 show the results for a surge motion of the platform with a misalignment of 45° with the wind direction and a reduced frequency of 0.6. In comparison to the surge case, the wind wave misalignment increases the strength of the vortex structures formed at the edges of the wake. Additionally, the vortex structures are shed in an alternating fashion on the two sides of the wake, due to the misalignment angle, similarly to the yaw motion cases. This difference is probably caused

by wake meandering, as the wake is deflected in the crosswise direction due to the misaligned motion of the turbine. Additionally, significant differences are observed at the centre of the wake, as the root vortices merge into stronger and larger structures. The formation of these structures wake is probably connected to the velocity oscillations observed at the center of the wake for wind-wave misalignment cases (see Sect. 4.2).

In conclusion, the analysis of the vortex structures in the wake provided further insight into the dynamics of the wake in the

760 investigated conditions. Results show that platform motion may lead to the formation of larger coherent structures in the far wake, which are connected to significant velocity oscillations. The formation of these structures is predominant when the reduced frequency is 0.6 and when meandering of the wake is introduced, either due to yaw motion of the platform or wind-wave misalignment. Among the numerical methods, the FVW shows similar results to the ALM simulations until about 2 or 3 D from the rotor, but differences arise as the distance is increased. In fact, after about 3D the main vortex structures interact

with one another and collapse into larger and weaker structures. Such non-linear interaction cannot be captured by the FVW approach, as the vortices are modelled and convected downstream with a Lagrangian approach, rather than by solving the Navier-Stokes equations.





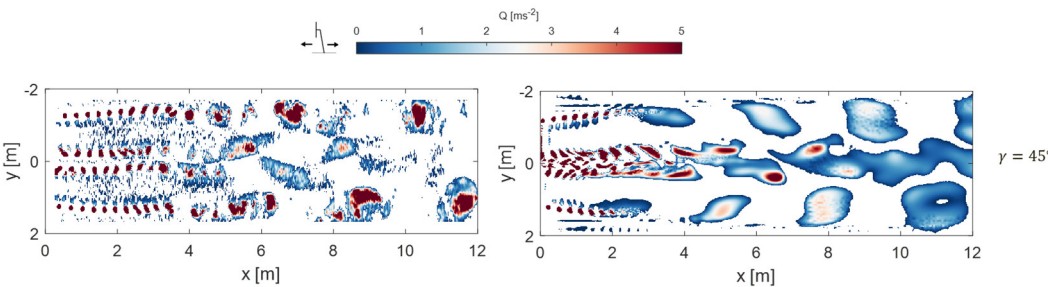

**Figure 30 Vortex structures in the wake of the wind turbine model under yaw motion with varying misalignment angles. Results obtained from CFD ALM simulations.**

## 5 Conclusions

The rotor and wake dynamics of a model wind turbine under imposed motion were further investigated as part of the NETTUNO project, leveraging additional operating conditions and wind tunnel measurements. The model response was investigated under sinusoidal surge, pitch, and yaw motions. Additionally, some cases characterized by a misalignment angle between the motion and the wind direction were studied, providing further insight into more complex operating conditions. The objective of the investigation was to evaluate the capabilities of engineering models, namely FVW and ALM CFD, to capture the rotor and wake response of a floating wind turbine. The numerical results were compared with experimental wind tunnel data complemented by high-fidelity additional data from blade-resolved and LES simulations.

The investigation of the rotor loads showed that FVW and ALM can correctly predict the rotor load oscillations in terms of thrust and torque over the considered range of conditions. By leveraging blade-resolved data a spanwise analysis of the loads was also carried out, showing that the FVW and CFD ALM correctly predict the aerodynamic response of the rotor, even though some differences may arise due to inaccuracies in the airfoil polars or tip effects. By comparing two different CFD ALM setups it was observed that the line averaged method the vortex method proposed by Sanvito et al. predict the angle of attack distribution with only minor differences, but the load distribution is affected by the employed regularization kernel, affecting the periodic oscillations in normal and tangential loads.

The capability of the FVW and ALM methods to capture the wake response up to 5D from the rotor was investigated.

The mean wake response was investigated in all the above-mentioned cases. Results showed that the average wake deficit shows only minor differences from a fixed-bottom configuration, in contrast to general consensus, which hypothesized a faster recovery of the wake. The increased mixing caused by platform motion causes increased turbulence levels in the wake, which may be detrimental to downstream machines leading to increased loading.

Among the simulation methods, the FVW approach can correctly capture the mean wake response in most operating conditions; however, results are impacted by the correct tuning of the main simulation parameters, namely the inflow turbulence, vortex viscosity and initial core radius. Such parameters are usually defined by the user following previous





experience, but the existence of experimental data can significantly improve the results obtained. In this perspective, the reader should always bear in mind that the comparison of the two engineering methods tested (FVW and ALM) is per se not completely fair, as the inputs needed for the models are different.

The CFD ALM model shows similar results in terms of mean wake deficit, and better agreement to the experiment at 5D from the rotor. Nevertheless, some differences from the experiment remain, which are connected to the limitations of the URANS approach in capturing the interaction between wake development and free-stream turbulence, as well as possibly tuning parameters of the ALM model, such as the regularization kernel. The unsteady wake response was then investigated, as a sinusoidal motion of the model may induce significant velocity oscillations in the wake at the frequency of platform motion. These oscillations are affected by the frequency and type of motion considered. All the motions that cause a deflection of the wake (pitch, yaw and wind-wave misalignment) cause increased velocity oscillations, in comparison to a pure surge motion of the wake. This difference is probably caused by wake meandering triggered by platform motion.

In terms of velocity oscillations, the FVW model correctly captures the wake dynamics up to about 3D from the rotor, even though the impact of motion on the onset of velocity oscillations is usually overestimated. At 5D from the rotor, the FVW does not solve the breakdown of the vortex structures leading to significant differences with the experiment. The CFD ALM model shows significant improvements in predicting wake velocity oscillations even at 5D from the rotor and correctly captures the significant increase in the velocity oscillations at a reduced frequency of 0.6, which is not captured by the FVW method.

Nevertheless, the ALM simulations are affected by the setup employed, as the ALM URANS simulations performed by UNIFI and POLIMI showed significant differences when predicting the amplitude velocity oscillations in the wake. The impact of nacelle, tower and robot, which are simulated by UNIFI but not included in the POLIMI setup, was investigated, but results showed only a minor impact on the wake response. Additionally, the kernel size was also excluded as the possible root cause for these differences, as additional tests performed by UNIFI matching the setup by POLIMI showed only minor differences to previous results. In general, this study showcases that the ALM methodology is a powerful tool; however, results may be impacted significantly by the employed simulation set up and careful validation and tuning campaigns are required.

Leveraging 2D velocity fields from previous ALM LES simulations under pitch motion, the results showed that ALM URANS approach might overestimate the formation of coherent structures in the wake, even though the URANS methodology captures the pulsation of the wake induced by the motion.

For yaw and wind-wave misalignment cases platform motion induces wake meandering in the wake, leading to a-symmetrical velocity oscillations in the wake. Under such operating conditions the FVW methodology showed significant differences with the experiment, especially at a distance of 5D from the rotor. Better results were obtained with the ALM URANS approach, even though the entity of the velocity oscillations might not be correctly captured.

Future work should focus on expanding the wake analysis to more realistic platform motion conditions, such as realistic motions involving all 6 DOFs. Additionally, the capability and limitations of the numerical methods presented here to predict the loading on downstream turbines should be evaluated, as these tools could provide significant insight for the design and development of floating offshore wind turbines.



**Data availability.**

Measurement data of the wind tunnel experiment are accessible at https://doi.org/10.5281/zenodo.13994980 (Fontanella et al., 2024). Data from numerical simulations and post-processing scripts available upon request.

**Authors contributions**

SC and AF carried out the numerical simulations. SC prepared the first draft and post-processed the results. FP and AB provided supervision to the numerical analysis. All the authors contributed to the analysis of the results and editing of the

manuscript. MB, AB, and VD procured the funding.

**Financial support**

This research has been funded by the European Union – NextGenerationEU, M4C2 I1.1, Progetto PRIN 2022 "NETTUNO", Prot. 2022PFLPHS, CUP D53D23003930006.

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
