# Peer review of "How accurately do engineering methods capture floating wind turbine performance and wake? A multi-fidelity perspective"

_Wind Energy Science, 2025_

## Author Comment (AC1)

Dear Editor,

We would like first to thank the Reviewers for the accurate and qualified observations. We truly appreciated the constructive criticism that made major improvements to the paper possible.

Based on their comments, an extensive revision of the work has been carried out. Our point-to-point responses have been highlighted in blue-colored text both in this communication and in the revised version of the paper. We really hope that this revised version can be now worthy of publication in *Wind Energy Science*.

Best regards,

*Alessandro Bianchini* on behalf of all the authors

ooo  ooo  ooo

**Reviewer #1**

The manuscript by Cioni et al. (10+ authors) presents a comparative study of numerical simulation methods for analyzing floating wind turbine wakes. Two established simulation approaches—Free Vortex Wake and ALM-URANS—are employed to generate wake data, which are then compared against experimental results up to 5 rotor diameters downstream of a model wind turbine with prescribed motion. The authors also utilize blade-resolved URANS and ALM-LES methods on two cases to provide additional insights. The comparison reveals varying levels of agreement and discrepancies between simulated and experimental wake characteristics.

Overall, this reviewer finds the manuscript valuable to the floating wind energy research community, and the experimental data presented is particularly interesting. However, the manuscript contains numerous errors, including typographical mistakes, incorrect citations, erroneous section numbers, and disordered figure arrangements. The overall quality falls short of standard expectations for a submitted manuscript and appears more akin to a first draft. Nevertheless, these issues can be addressed through careful revision.

**Given the number of errors identified, I hope the editor allows me to provided direct annotations on the PDF file of the preprint, with 70+ comments in total.** I request that the authors carefully consider each annotation, provide point-by-point responses, and resubmit the manuscript for a second round of review.

The authors would like to thank the Reviewer for his/her careful evaluation of the manuscript and constructive criticism. We have addressed all the relevant comments and checked the new manuscript in depth to try eliminating the typographical mistakes of the first draft. We have marked all the relevant changes in the manuscript in blue and provided a point-by-point response to the Reviewer questions (RC) in this document. Any reference to Sections, Figure or line numbers have been included in this document referring to the revised version of the manuscript. Our answers in this document (AC) have also been highlighted in *blue*, and we have provided appropriate references to the text.

RC: Engineering method is a quite vague term. In this paper, different numerical simulation methods are compared with experiment. Please modify engineering methods to numerical simulation methods.
AC: The Reviewer's comment is relevant. While using experiments as a benchmark, our study is fully based on "numerical simulations". Such term, however, does not distinguish between those which solve the blade-flow interaction (like blade-resolved CFD) and those that are based on "engineering" modeling, like the use of tabulated airfoil polars as input data to estimate blade forces or a number of semi-empirical tuning constants (like those for wakes in the FVW method). These latter methods are the main subject of the study, and that is why we are proposing the use of "engineering simulation methods" (often used in literature). Based on your comment, we have better introduced such concept in the study.

RC: It is not clear what a multi-fidelity perspective references to . Maybe modfy to "A comparison between different simulation fidelity with experimental data"

AC: Thank you for your comment. We agree that the original title of the study could be improved to highlight that multi-fidelity refers to numerical simulations. We have changed the second part of the title to: "A critical perspective based on multi-fidelity simulations".

RC, line 19: Typo
AC: We have corrected the manuscript, thank you.

RC, line 23: What is a platform reduced frequency, please define. Later in the manuscript, it is found to be the rotor reduced frequency. Please make them consistent.
AC: The platform reduced frequency refers here to the rotor reduced frequency, as defined in the following parts of the manuscript and pointed out by the Reviewer. We have modified the manuscript to guarantee consistency across the text. Additionally, we have modified the abstract to avoid confusion about the definition of the reduced frequency.

RC, line 31: Please cite the final published version.
AC: Thank you for your comment. We have modified the reference to this paper by citing the final version.

RC: line 52: Dear authors, please don't describe previous studies as preliminar results. They are outcomes of serious studies and have been published in peer reviewed journals. They do not show conflicting results. They just show the wake will behave differently when the motion and environment condition are different. This difference is well documented in the literature.
AC: Thank you for your comment. We agree that the use of the term "preliminary results" for published works is improper (it is somehow due to a "false friend" in Italian – the idea was to mainly refer to something before in time) and we have modified the manuscript accordingly. Additionally, we have rephrased our comment concerning the differences observed across the currently published results, highlighting the impact of different testing and environmental conditions.

RC, line 54: Please define the abbreviation "NETTUNO "
AC: The term NETTUNO is not an abbreviation but represents the name of the research project. To further clarify the reference to the research project, we have also included a reference to the initial paper, which includes the experimental results.

RC, line 68:The author are citing a wrong reference. In Kleine et al. , the lowest frequency considered is equal to 0.5 times the rotor frequency, which is relatively high. For low frequency oscillations, Li et al. [1] found the wake meandering can be triggered by platform side-to-side motion with 0.2<fr<0.6, which is more appropriate reference. [1] Li, Z., Dong, G. and Yang, X. (2022) 'Onset of wake meandering for a floating offshore wind turbine under side-to-side motion', Journal of Fluid Mechanics, 934, p. A29. doi:10.1017/jfm.2021.1147.
AC: We agree that the motions considered in the reference are not low-frequency. With this reference we wanted to discuss the onset of velocity oscillations at the frequency of platform motion. The text has been corrected accordingly. Additionally, we consider the suggested reference to be valuable for the discussion and for this reason we have added it to the manuscript.

RC, line 69: Please cite the final version published in 2023, instead of this discussion version (preprint).
AC: Thank you for your comment. We have modified the reference.

RC, line 77: What is the first issue already? Please use subsection titles to better structure the text.
AC: Splitting the introduction into subsections is often discouraged by reviewers and editors; also, the introduction here is not very long. However, upon double-checking the paper, we agree with the Reviewer that the Various paragraphs in the introduction could have been tied together in a more coherent manner. For this reason, we have made some adjustments in the manuscript to improve readability.

RC, line 99: "surge, pitch,"
AC: Thank you for your comment. We have added the missing motion cases to the list.
RC, line 100: Please add references to support this claim, or remove it.
AC: We have further clarified our comment, specifying that this work represents the most extensive comparison of engineering models with available experimental data. We have also included relevant references to support this claim.

RC, line 103: Please add reference. Explain properly what is a multi-fidelity analysis?
AC: We have added a reference to the relevant study. In the manuscript, when referring to a multi-fidelity analysis we are referring to the use of different stand-alone numerical methods with varying degrees of fidelity. By employing such different models, we can highlight their limitations and capabilities, providing valuable insight for future research. In response to Reviewer's right comment, we have added a brief description of what we refer to when writing about multi-fidelity analysis in the original manuscript.

RC, line 103: It will be very useful to compare the computational cost between ALM-LES and ALM-URANS, and BR-URANS, please add it in a proper place.

AC: We agree that adding further details about the computational cost of the different methods can be valuable for the reader. We have added a comment for each methodology in Sect. 3, where the numerical models are described. In particular, lines 238, 316, and 319 provide information about the computational cost of the FVW, ALM URANS, and ALM LES simulations, respectively. For the blade-resolved simulation the same details are provided in line 351.

RC, line 118: Is this scale correct?

AC: Sorry for the typo, the scaled model is 1:75 in comparison to the DTU 10MW. The text has been corrected.

RC, line 121: Please justify it, or remove this claim.

AC: We have removed the comment from the manuscript as it was indeed too vague.

RC, line 125: It is different from the "platform reduced frequency" in the abstract. Please modify the abstract. This is reduced frequency is the same as the definition of Strouhal number in fluid mechanics.

AC: Thank you for your comment. We have modified the abstract to maintain consistency across the text. The definition of the rotor-reduced frequency is indeed the same as the one of the Strouhal number.

RC, line128: The free-stream velocity should be defined.

AC: We agree that the inflow velocity should be defined before providing details about the reduced frequency. We have added the inflow speed value to the manuscript.

RC, line 130: You need to assume a U at full scale to do this calculation, please provide this value.

AC: We agree that, in order to connect a reduced frequency value to a dimensional frequency at full-scale, an inflow speed should be defined. For this reason, we have rephrased the manuscript to express the comparison between full-scale and model frequency in dimensional terms.

RC, line 133: There are a number of previous studies showing dynamic yaw motion can lead to wake meandering, you don't really need to test this possibility.

[1] Munters, W., & Meyers, J. (2018). Dynamic strategies for yaw and induction control of wind farms based on large-eddy simulation and optimization. Energies, 11(1), 177.

AC: We agree with the Reviewer that previous work has investigated strategies for dynamic yaw control in fixed-bottom machines, showing the onset of wake meandering. In such conditions, it is indeed not necessary to evaluate how yawing of the turbine might impact wake dynamics. However, in this work, the yaw motion of the turbine is induced by platform motion and not by an active yaw control. This is the case, for example, of floating cases with severe wind-wave misalignment. Moreover, the different origins for yaw motion will lead to differences in operating amplitudes and frequencies. In this work, we investigated operating conditions representative of FOWTs, where it is still not clear how and to which extent yaw motion may impact wake dynamics. For example, larger amplitudes and lower frequencies of motion are usually considered in dynamic yaw control cases. We have modified the manuscript highlighting previous work on fixed-bottom machines and showcased the differences for FOWTs.

RC, line 144: sway and roll motions are not shown in figure 1 nor in table 2.

AC: The authors agree that pure sway and roll motions are not shown both in Figure 1 and Table 2. These motions are not reported in both the table and the figure because during the tests pure sway and roll motions have not been investigated. Instead, only cases characterized by a combination of surge+ sway and pitch + roll were tested. The combination of surge and sway is reported in Figure 1; however, these cases are reported in Table 2 as surge motion cases and a corresponding misalignment angle. To avoid any confusion, we have divided the misalignment cases in Table 2 as surge+ sway cases. Pitch and roll cases were tested during the wind tunnel campaign, but they are not reported in the manuscript due to similarities with surge and sway operating conditions. To avoid again confusion, we have eliminated pitch and roll cases from the list of cases.

RC, line 173: The logic here is difficult to understand? Why use a case with largest difference ? Are simulations refer to FVM and ALM simulations?

AC: In this section of the manuscript we want to explain the choice of the operating conditions, which were simulated using a blade-resolved approach. In fact, due to the significant numerical cost of the methodology, only a single test case could be simulated. In that study, among the available operating conditions of the NETTUNO project, the pitch motion at an amplitude of A=0.3° and f=2Hz was selected. The choice was made following the results obtained by participants of OC6 phase III project, where it was observed that both ALM and FVW results showed the most significant differences with the experiment. In fact, by performing blade-resolved simulations we wanted to further investigate this test case and try to explain the source of the observed differences. In particular, we hypothesized that the discrepancies were connected to the onset of unsteady aerodynamic effects, which are not captured by the ALM and FVW methods. By performing blade-resolved simulations we were able to provide a benchmark for the two methods and to investigate the spanwise rotor-aerodynamic response. We have added a comment to the manuscript to further explain this decision.

RC: Do you aware that this similar sentence appears three times in this manuscript?
AC: Thank you for your comment. We have now deleted this comment from the manuscript as we agree that a similar sentence was already present in other sections of the text.

RC, Table 2, caption: It will be useful to normalize the motion amplitude by diameter.
AC: Thank you for your comment. We have normalized the amplitude by the rotor diameter.

RC, line 201: timeseries -> time series
AC: Thank you for your comment, we have modified the manuscript.

RC, line 210: It is not very clear which condition is employed to generate figures 3, 4, 5.  3.1.1 Tuning the convection velocity
AC: We agree that the operating conditions used for Figures 3, 4 and 5 were not well defined and limited to the captions of the figures. We have now added a description in the manuscript to include further details. We have also modified the subsections of this chapter, as suggested.

RC, line 240: 3.1.2 Tuning  simulation parameter
AC: We have added the subsection to improve the structure of the text.

RC, equation 3: The two terms in the denominator have inconsistent dimensions
AC: Thank you for your comment. We have corrected the typo in the equation.

RC, line 263: what are the most relevant cases?
AC: The subset of cases simulated using the ALM approach is reported in Table 2. We have clarified the manuscript to highlight where the reader can find detailed information about all the simulated operating conditions.

RC, line 266: Which blind test? what is the relation between this blind test with the present work? Provide information, or remove this sentence.
AC: We agree with the Reviewer that this sentence was not clear. By "blind test" we wanted to refer to the methodology employed in comparing the results of the two ALM methodologies from UNIFI and POLIMI. In particular, we wanted to highlight that the simulations were performed without a prior tuning campaign of the methodologies which could have led to improvements in the agreement between the two simulation setups. We have modified the manuscript to clarify this point.

RC, line 358: what is the unit of this time step?
AC: Thank you for your comment, the unit time is seconds. We have added it to the manuscript.

RC, line 309: what is a successful cycle?
AC: This was a typo. We wanted to write "subsequent" to highlight the fact that the URANS simulations showed only minor differences between subsequent cycles of motion.

RC, line 348: upwind is for spatial discretization scheme, how to employ a upwind scheme in time?
AC: Thank you for your comment. Indeed, this is a second order implicit in time. We have corrected the manuscript.

RC, Section 4: This is section 4.
AC: We have corrected all the Section numbers in the manuscript.

RC, line 353: I don't think they are engineering models. They are numerical simulations.
AC: Please refer to one of the first comments for more explanations on this choice of wording. However, the Reviewer is right in saying that this specific section was not clear; it has been now rephrased.

RC, line 374: With two points, you cannot have demonstrate a linear trend.
AC: We agree that with only two points a linear trend cannot be demonstrated. For this reason, we have rephrased the manuscript to highlight that this conclusion is reached only from the operating amplitudes where data is available for three reduced frequencies.

RC, Figure 7: What is the difference between the upper left and the upper right figures?  I found them identical.  Please explain them in the caption.
AC: The Reviewer is perfectly right. Due to an error in the post-processing, the same results were plotted for both surge and pitch. We have corrected the figure accordingly.

RC, figure 8: For the top figure, showing the origin i.e., (0,0) is suggested.
AC: We agree with the Reviewer, and we have modified the Figure accordingly.

RC, line 393: Here, the behavior of yaw induced velocity is quite interesting, but the discussion on it is very limited. Please provide more discussion on this dependency of Mz on fr. It is interesting to see that Mz increases with fr while the yaw amplitude is fixed. The present numerical and experimental results may be employed to improve fast predicting engineering models for floating wind turbine wakes, such as the resolvent-based motion-to-wake model proposed by Li and Yang [1]. In their work, rotor force variation is modelled in a quasi-steady way.
[1] Li, Z. and Yang, X. (2024) 'Resolvent-based motion-to-wake modelling of wind turbine wakes under dynamic rotor motion', Journal of Fluid Mechanics, 980, p. A48. doi:10.1017/jfm.2023.1097
AC: We agree with the Reviewer that the present result may be provide a valuable benchmark for the tuning of future engineering models. The increase of the amplitude of the oscillations of the momentum around the z-axis as a function of the reduced frequency while keeping the amplitude of motion constant is indeed quite interesting. While an in-depth investigation of such trend is out of the scope of this work, we believe that such trend is connected to the increase in relative velocity induced by the motion. In fact, when the reduced frequency is increased while keeping the amplitude of motion constant, the apparent maximum velocity induced by the motion is also increased. Additional discussion has been added to the paper.

RC, line 409: typo
AC: Corrected.
RC, line 424: Figure 10
AC: Corrected.

RC, line 434: I am not sure if this conjure is correct. Because the airfoil polar are the basic input for ALM. How do you explain the good comparison with experiment?
AC: The difference among ALM and FVW and the BR simulation in terms of oscillation amplitude at about 70% of the span was explained by the differences in the spanwise load distribution observed in Fig. 9, where some differences can be observed at the same spanwise location. As this difference is observed for all the lifting line models, we hypothesize that the difference itself is connected to small discrepancies between the airfoil polars employed in these models and the 3D aerodynamic response observed in the blade-resolved CFD. This also bases on a long experience of the authors with such methods. As differences between the models are localized between 60 and 80% of the span, the impact on the rotor integral performance (for example in terms of rotor thrust and torque) remains limited. We have added a comment to further clarify this point.

RC, line 440: It is very strange to see the oscillation amplitude show such a difference between the blades. Please elaborate.
AC: The Reviewer's comment is pertinent. The differences observed across the three rotor blades in terms of amplitude of the normal load oscillations is due to the pitch motion of the rotor and the selected combination of rotor rotational frequency and platform motion frequency. In fact, as the rotor rotational speed is two time the frequency of platform motion, each blade is found always in the same azimuthal position for every cycle of motion. Hence, for each blade, the impact of the asymmetry introduced by pitch motion can be clearly observed in the oscillations of the normal loads. In fact, depending on the azimuthal position of each blade during a cycle of motion, pitch motion alters the local AoA both due to the changes in relative velocity (caused by the changes in structural velocity of the turbine) and due to the misalignment of the rotor with respect to the inflow. In this specific case, the two contributions add up for blade 1, resulting in an increase in the rotor load oscillations. In contrast, the two contributions compensate for the remaining two blades. We have added a brief description of this result in the manuscript to guarantee that the discussion is self-contained. However, an in-depth description of this effect is beyond the scope of this work. For this reason, the interested reader is invited to read the original publication, provided as a reference, where these results have been discussed.

RC, Sect. 5: This should be section 5.
AC: Thank you for your comment. We have corrected the section numbers across the manuscript.

RC, line 475: in the near wake.
AC: Changed.

RC, line 500: twisted sentence. Please rephrase.
AC: We have simplified the sentence to improve clarity.

RC: Figure 14 is about the mean flow, but this comment is on the turbulence intensity. Is it referring to figure 15?
AC: In this sentence we are referring to the turbulence intensity profile at 3D from the rotor, corresponding to Fig. 13 rather than 14. We have corrected the manuscript accordingly.

RC, line 527: Please comment this conjure using the data obtained from experiment and simulations from this study.
AC: Thank you for your comment. We have simplified the manuscript in this section, as the discussion was not directly linked to the findings of this study.

RC, line 554: Figure 19 appears before 17 and 18.  I don't think it is correct. Figure 19 compares surge and pitch.
AC: The text actually refers to Figure 17. Thank you for your comment, we have modified the manuscript accordingly.

RC, line 572: It is not clear which figure support this statement.
AC: In this final part of this Section we want to provide a final summary of the results to improve the readability of the paper. This comment refers to the previous results shown within this Section. We have modified the manuscript to further clarify this point.

RC, line 584: Figure 18?
AC: Thank you for your comment, we have modified the reference to the Figure.

RC, Figure 18: This should be the oscillation part of wake deficit. Since the time-averaged deficit should not be zero.
AC: Thanks for pointing this out. Indeed, the figure represents the oscillation of the wake deficit. We have modified the caption of the figure accordingly.

RC, line 602: Please elaborate such a different behavior.
AC: In terms of wake deficit oscillations, the surge motion results obtained for a platform motion amplitude of 0.016 m are the only ones that show a linear increase of the amplitude with increased reduced-frequency. In contrast, for pitch motion cases and increased amplitudes of surge motion the results show a reduction in the wake deficit oscillations at $f_r = 1.2$, and no linear increase is observed. The results observed for surge motion might indicate that the model is operating in quasi-steady conditions for small amplitudes of surge, due to the small apparent velocity induced by the motion. We have clarified this point in the manuscript.

RC, line 638: Figure 20, right column?
AC: Correct. We have modified the manuscript accordingly.

RC, figure 20: Please use the diameter to normalize y and the motion period to normalize time.
AC, line 634: Thank you for your comment. We have modified the figures by normalizing the corresponding variables.

RC, line 658: figure 21
AC: Changed.

RC: line 671: Figure 22
AC: This comment refers to Figure 23. We have modified the manuscript accordingly.

RC, Figure 23: Please specify the reduced frequency. The font size of the colorbar and title of subplots should be increased.
AC: Thank you for the suggestion. We have modified the figure and the corresponding caption.

RC, Figure 24: The TIs are very different. Are they defined in the same way? Please specify how the TIs are computed.
AC: Due to an issue in the colormaps of the URANS results, the differences between the two methods were amplified. We have updated the plot accordingly. Additionally, we have modified the colormap to guarantee consistency across the manuscript, as suggested by another reviewer. To avoid any confusion, we have also introduced the definition of the TI in the manuscript in Eqs. 2 and 5.

RC, line 690: in Figure 23
AC: Corrected.

RC, line 692: Please elaborate this phenomenon.
AC: In the yaw motion cases, the experiments show some significant differences in wake dynamics when moving from 3D to 5D downstream the rotor. In fact, at 3D downstream, the wake oscillates in the crosswind direction at the same frequency of platform motion. Such lateral oscillations correspond to anti-symmetrical oscillations observed in Figure 22. In fact, as the wake moves to one side, the streamwise velocity is reduced on that side due to the presence of the wake, while on the opposite side the streamwise velocity increases, since the wake has moved away. At 5D from the rotor a similar trend in the velocity oscillations is not observed. In contrast, the experimental results show homogeneous velocity oscillations across the whole horizontal traverse shown in Figure 23. This result suggests that the wake is no longer oscillating in the crosswind direction but is pulsating at the frequency of motion, with subsequent increases and decreases

in the wake streamwise velocity. Based on the reviewer's pertinent comment, we have modified the manuscript to further clarify the different wake dynamics (line 693).

RC, line 697: I don't see it is a closer distribution. They are very different.
AC: While there are still some significant differences between the ALM results and the experiment, the results show better agreement in comparison to the FVW method, as the wake seems to be transitioning to a full wake pulsation, closer to the experiment. We have clarified the manuscript to highlight that differences remain between the ALM results and the experiment.

RC, Figure 26: Please specify the angle of misalignment, amplitude of surge motion, etc...
AC: Thank you for your comment, we have added the additional details to the caption.

RC: This sentence appears three times already.
AC: We have modified the manuscript in the remaining parts to avoid repetitions.

RC, figure 28: The contour of LES is much better looking. But if the same colorbar is employed, blue color should also appear in the contour, since it represent Q = 0. Please verify the plot.
AC: Thank you for your comment. We have double-checked the plot, and we have observed that indeed the blue color should appear also in the LES plot. The difference was due to a mistake in the post-processing which led to a wrong application of the colorbar. The relevant plot has been corrected. Additionally, following the comments from another Reviewer, we have changed the colormap to maintain consistency across the manuscript.

RC: and ALM simulations
AC: Thank you for your comment. We have modified the manuscript accordingly.

RC, line 835: This statement is wrong. The NS equation introduces nonlinearity only by the convection term. This is also included in the FVM if the mutual induction is considered.
AC: Thank you for your comment. We agree that the FVW method accounts for non-linear interactions through mutual induction. With this sentence, we wanted to highlight the limitations of the FVW approach in capturing the viscous interactions between vortex filaments. In fact, this method accounts for viscous effects through semi-empirical correlations (i.e., the velocity profile induced by each filament is computed using a vortex model and tuned with the vortex viscosity parameter $\delta_v$). We have corrected the manuscript accordingly.

RC, Figure 30: there is no wave-wind misalignment for the cases with yaw motion, see table 2. Is surge with misalignement? Please specify the misalignment angle etc. I think results are obtained form FVM and ALM.
AC: Results were indeed FVW and ALM. We have modified the figure accordingly and added further details to the caption.

RC, Section 6: The conclusion should be written in a more concise and structured way. Use some bullet to make your points and guide the readers.
It is suggested to compare the strength and limitation of FVM and ALM CFD directly. I don't know if you need to include two-different ALM methods. I am feeling that they provide similar results.
AC: Thank you for your comment. We have modified the conclusions of the paper extensively to highlight the outcomes of the study in terms of strengths and limitations of the different numerical models employed. Additionally, we have clarified the advantage of presenting different ALM setups in this work.

RC Conclusions: Simulation methods
AC: Thank you for noting. We have modified the conclusions accordingly.

RC, Conclusions: Please replace ALM CFD with ALM URANS everywhere.
AC: Done.

RC, Conclusions : what do you define high-fidelity data? In wind energy, the high-fidielity simulaiton often refered to large eddy simulation, when compared to RANS/URANS. Under this criteria, the blade resolve URANS is not a high-fidelity method.
AC: While high-fidelity data might be commonly referred to LES simulations, we believe that the additional level of fidelity provided by a blade-resolved approach can indeed be referred to as "high-fidelity". In fact, a significant improvement is achieved in the modeling of fluid-rotor interactions in comparison to an ALM approach, which still requires predefined airfoil polars and semi-empirical corrections. Nevertheless, to avoid any confusion for the reader, we have modified the manuscript to avoid referring to blade-resolved simulations as high-fidelity.

RC, Conclusions: As the wake is only investitgatd upto 5D, the far wake recovery as revealed by the "general consesus" is not known.

AC: Thank you for your comment. We have substantially modified the conclusions of the paper to avoid confusion and improve clarity in the definition of the results.

---

## Author Comment (AC2)

Firenze, 20/01/2026

Dear Editor,

We would like first to thank the Reviewers for the accurate and qualified observations. We truly appreciated the constructive criticism that made major improvements to the paper possible.

Based on their comments, an extensive revision of the work has been carried out. Our point-to-point responses have been highlighted in blue-colored text both in this communication and in the revised version of the paper. We really hope that this revised version can be now worthy of publication in *Wind Energy Science*.

Best regards,

*Alessandro Bianchini* on behalf of all the authors

ooo   ooo   ooo

**Reviewer 2**

In the manuscript, the authors conduct a systematic comparison of engineering-fidelity models for a scaled floating wind turbine subject to a variety of platform motions. Models of various fidelity are compared with higher-fidelity approaches (some LES) as well as experimental measurements. Considered platform motions principally include surge, pitch, and yaw with some misaligned conditions also considered. The comparisons are comprehensive and will be a valuable resource to the community in guiding the selection of appropriate approaches. The basic conclusion is that FVM and ALM are capable of correctly predicting loads and the steady and unsteady wake responses under surge and pitch conditions. For yaw and misalignment, ALM is superior to FVM.

The authors would like to thank the Reviewer for his/her careful evaluation of the manuscript and the constructive criticism. We have addressed all the comments you provided, and we have marked the changes in the manuscript in *blue,* providing a point-by-point response to all the Reviewer's questions (RC) in this document. Any reference to Sections, Figure or line numbers that have been included in this document do refer to the revised version of the manuscript. Our answers in this document (AC) have also been highlighted in *blue*.

RC2: Not all of the figures are referenced in the text, and some of the figure references are incorrect.
AC: Thank you for your comment. We have extensively reviewed the manuscript and corrected any typographical errors.

2. The results are presented in a mix of dimension and non-dimensional units. Considering the scale, presenting results in non-dimensional form will aid in generalizing to full-scale conditions.
We agree that normalizing the main parameters would improve the generalization of the paper. We have modified the manuscript accordingly.

---

## Author Comment (AC3)

Firenze, 20/01/2026

Dear Editor,

We would like first to thank the Reviewers for the accurate and qualified observations. We truly appreciated the constructive criticism that made major improvements to the paper possible.

Based on their comments, an extensive revision of the work has been carried out. Our point-to-point responses have been highlighted in blue-colored text both in this communication and in the revised version of the paper. We really hope that this revised version can be now worthy of publication in *Wind Energy Science*.

Best regards,

*Alessandro Bianchini* on behalf of all the authors

ooo   ooo   ooo

**Reviewer #3**

The topic of floating WT's is an important topic, and the availability of experimental wind turbine data under simulated floating offshore platform movement is indispensable. The comparison with several numerical simulation methods gives a good impression how this experimental data can be used for code validation.

The authors would like to thank the Reviewer for his/her appreciation of the study and for the constructive criticism.

RC: The article in its current form contains many textual errors (in text, captions, references).

AC: Thank you for your comment. We have carried out an extensive review of the manuscript to correct any typographical errors we could identify.

RC: The figures would benefit from using a uniform layout and color scheme.

AC: We agree with the Reviewer that the visualization of some figures could be improved by maintaining a uniform style across the manuscript. We have modified the figures in the paper accordingly.

RC: This reviewer struggled to identify the goal of the article: was it to demonstrate the usefulness of the experimental data or was it to validate the several numerical implementations? (I guess it's the former) This uncertainty is also present in the title of the article that is formulated as a question, that can be answered with "It depends...".

AC: Thank you for this criticism, which helped us re-thinking the way to present our study. The main objective of the paper was to compare the different numerical methods with the experimental data and to evaluate their capabilities and limitations, so as to provide the reader an overview of what he/she can expect when making using of such methods in simulating FOWTs. We have modified the manuscript, especially in the introduction and conclusions, to clarify the objective of this work and to further highlight the comparison between numerical models and experimental data.

RC: The results of the simulations were from specific implementations of specific flow modelling choices, using specific runtime settings. The discussion of the results should make clearer that the conclusions are valid for these specific computer programs using specific settings and cannot be generalized to, for example, "the FVW and ALM methods ...".

AC: We agree with the Reviewer that the results reported here are valid only for the specific simulation employed and that different codes or simulation strategies may perform differently in comparison to the experiments. For the ALM models, we have tried to provide a more general outlook by comparing two different implementations and numerical setups, which have not undergone preliminary tuning or comparison, even though this remains a limited sample of all possible implementations and setup choices. We have now tried to better highlight this limitation of the study in the Conclusions, clarifying that the outcomes are limited to the specific implementations tested and further work is required to achieve a more general validation of the numerical models.